# Order from Chaos: Physical World Understanding from Glitchy Gameplay Videos

Meng Cao[1][*], Haoran Tang[2][*], Haoze Zhao[1][*], Mingfei Han[1], Ruyang Liu[2], Qiang Sun[1,3][†],
Xiaojun Chang[1], Ian Reid[1], Xiaodan Liang[1,5][†]
[1]Mohamed bin Zayed University of Artificial Intelligence  [2]Peking University
[3]University of Waterloo  [4]University of Toronto  [5]Sun Yat-sen University
[*]Authors contributed equally to this research.  [†]Corresponding author.

Reviewed on OpenReview: https://openreview.net/forum?id=Oe5TdpPv1b

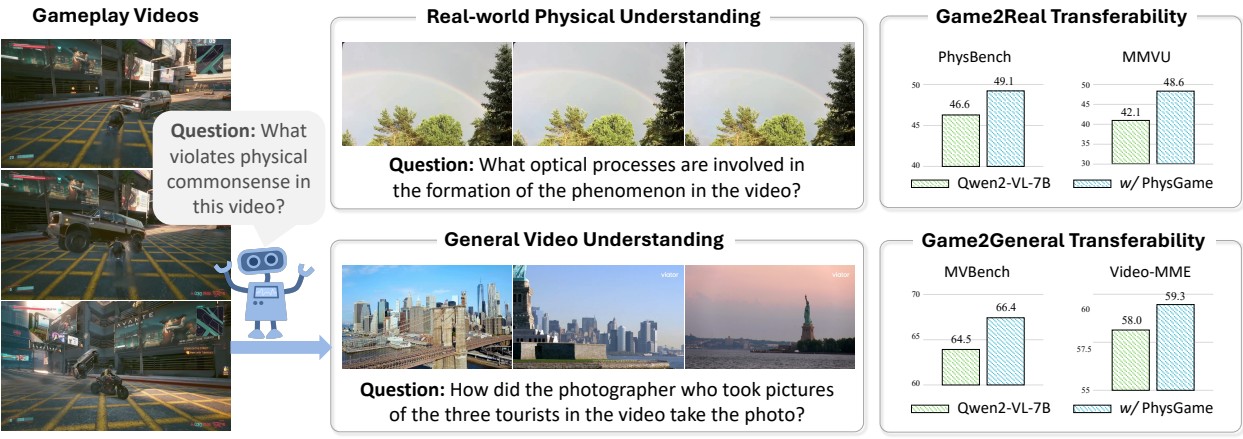

Figure 1: We propose PhysGame, an instruction-tuning dataset constructed from gameplay videos with glitch-induced question-answer pairs. Our method demonstrates consistent performance improvements on downstream benchmarks for both real-world physical understanding (Game2Real) and general video understanding (Game2General) tasks.

## Abstract

Understanding the physical world, including object dynamics, material properties, and causal interactions, remains a core challenge in artificial intelligence. Although recent multi-modal large language models (MLLMs) have demonstrated impressive general reasoning capabilities, they still fall short of achieving human-level understanding of physical principles. Existing datasets for physical reasoning either rely on real-world videos, which incur high annotation costs, or on synthetic simulations, which suffer from limited realism and diversity. In this paper, we propose a novel paradigm that leverages glitches in gameplay videos, referring to visual anomalies that violate predefined physical laws, as a rich and scalable supervision source for physical world understanding. We introduce PhysGame, an instruction-tuning dataset containing 140,057 glitch-centric question–answer pairs across five physical domains and sixteen fine-grained categories. To ensure data accuracy, we design a meta-information–guided prompting strategy that utilizes gameplay metadata such as titles and descriptions to guide high-quality QA generation. Complementing PhysGame, we construct GameBench, an expert-annotated benchmark with 880 glitch-identified gameplay videos designed to evaluate physical reasoning capabilities. Extensive experiments show that PhysGame significantly enhances both Game2Real transferability, improving the real-world physical reasoning performance of Qwen2.5-VL by 2.5% on PhysBench, and Game2General

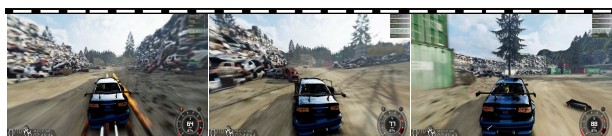 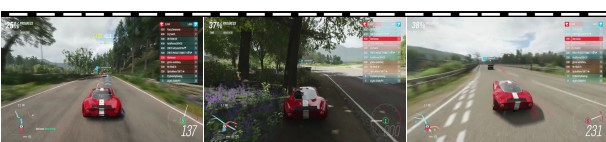

***Question:*** Is there any notable glitch in the video that affects the gameplay?
***Answer:*** While navigating the course, the car door is oddly hanging open, which does not seem to affect its speed or performance. This could be considered a minor glitch, as such damage should typically impact aerodynamics or handling in a realistic scenario. There is no major collision effect noticeable when cars collide, suggesting the physics might not fully replicate real-world damage

***Question:*** How would you describe the glitch in the video?
***Option:***
A: Blocked by a tree, the car's speed instantly surged from zero to over 200 mph.
B: While the player's car is on the bridge, it casts no shadow.
C: On the left lane of the tree-lined road, there are flickering red light patches.
D: The clouds in the sky exhibit an unusual green hue.

Figure 2: **Sampled examples** in the proposed PhysGame instructing dataset (left) and the GameBench evaluation benchmark (right).

transferability, yielding a 1.9% gain on the MVBench benchmark. Moreover, PhysGame-tuned models achieve a 3.7% absolute improvement on GameBench, demonstrating enhanced robustness in detecting physical implausibilities. These results indicate that learning from gameplay anomalies offers a scalable and effective pathway toward advancing physical world understanding in multimodal intelligence.

# 1 Introduction

> "In all chaos there is a cosmos, in all disorder a secret order."
> — Carl Jung, *Archetypes and the Collective Unconscious*

Physical world understanding McCloskey et al. (1983); Melnik et al. (2024); Duan et al. (2022) represents a fundamental challenge in artificial intelligence, encompassing the understanding of physical properties, object interactions, and temporal dynamics. Even before language acquisition, children begin to grasp fundamental principles of the physical world by observing the environment around them Hespos & Spelke (2004). While recent multi-modal large language models (MLLMs) Achiam et al. (2023); Reid et al. (2024); Brown et al. (2020); Touvron et al. (2023) have achieved remarkable advancements across various domains, their capabilities in physical world understanding remain substantially inferior to human-level intelligence Wang et al. (2024c); Zheng et al. (2024); Chow et al. (2025); Zhao et al. (2025); Xiang et al. (2025). For example, on the physical understanding benchmark PhysBench Chow et al. (2025), the state-of-the-art MLLM GPT-4o OpenAI (2024) achieves merely 49.49% average accuracy, falling far short of the human-level performance (95.87%).

The majority of existing physical world understanding datasets can be categorized into two types: *1) Real-world video data* Chow et al. (2025); Zhao et al. (2025); Xiang et al. (2025): While capable of capturing complex backgrounds and heterogeneous compositional elements for physical reasoning, such data typically requires high annotation costs involving careful curation of specific video types and time-consuming question-answering (QA) annotation processes; *2) Simulator-generated synthetic data* Yi et al. (2020); Zheng et al. (2024); Chen et al. (2022); Bear et al. (2021): These datasets employ physical simulators (*e.g.*, Blender Conlan (2017) and Unity Haas (2014)) to create controllable scenarios for physical reasoning. However, they often exhibit uniform backgrounds and simplistic visual primitives (*e.g.*, cube, cylinder), leading to a substantial domain gap between synthetic representations and complex real-world physical scenarios Wang et al. (2024c). Based on the preceding analysis, we contend that neither data type is well-suited as a source for generating *large-scale instruction-tuning datasets* aimed at physical world understanding.

In this paper, we propose an alternative perspective by learning from glitches in gameplay videos Chen et al. (2021); Ling et al. (2020); Nantes et al. (2008), *i.e.*, inconsistencies between pre-defined physical rules and observed visual manifestations caused by rendering errors, physics engine limitations, or unexpected object interactions. For example, Figure 1 illustrates an implausible scenario in which a motorcycle unrealistically propels a car into the air, violating basic physical laws. Our motivation is inspired by the *philosophical*

*notion of Order from Chaos*, which emphasizes that understanding often emerges more clearly by identifying and characterizing deviations from expected regularities. In other words, explicitly modeling violations of physical rules enables a sharper grasp of the principles that normally govern the physical world. To this end, we introduce **PhysGame**, an instruction-tuning dataset constructed by harvesting **game**play videos with glitches to advance **phy**sical world understanding. PhysGame contains 140,057 question–answer pairs targeting glitch-related content across five key physical domains (*i.e.*, mechanics, optics, material properties, thermodynamics, and electromagnetism) and sixteen fine-grained categories (*e.g.*, gravity and velocity). To ensure the accuracy of the generated QA pairs, we propose a meta-information–guided prompting strategy, which incorporates video-associated metadata (*e.g.*, content-summarizing video titles) when prompting leading MLLMs such as GPT-4o OpenAI (2024). Furthermore, we introduce **GameBench**, a complementary evaluation benchmark comprising 880 glitch-identified gameplay videos, each accompanied by expert-curated multiple-choice questions that explicitly probe glitch characteristics. The sampled examples of PhysGame instruction dataset and GameBench evaluation benchmark are presented in Figure 2. Compared to existing approaches, our proposed PhysGame captures dynamic and diverse scenes that resemble real-world physical complexity, while remaining highly scalable due to the abundance of online data and reduced annotation cost through glitch identification. In addition, incorporating previously unseen glitches from the PhysGame dataset during post-training further enhances generalization and remedies the deficiencies of current MLLMs in glitch identification within gameplay videos. For example, built upon Qwen2-VL Wang et al. (2024a), post-training with the PhysGame dataset yields an absolute improvement of 6.3% in average accuracy on our constructed GameBench benchmark (*cf*. Table 3).

In terms of the aforementioned advantages, we demonstrate that PhysGame consistently enhances downstream benchmark performance across both physical understanding and general video understanding tasks. Specifically, two transferability characteristics of PhysGame are highlighted: 1) **Game2Real** Transferability: Despite being constructed from *simulated* gameplay videos, PhysGame finetuning significantly improves *real-world* physical reasoning. As shown in Figure 1, Qwen2-VL Bai et al. (2025) achieves 2.5% absolute improvement (46.6% *vs.* 49.1%) on PhysBench benchmark Chow et al. (2025) after finetuning on the proposed PhysGame dataset; 2) **Game2General**: Physical knowledge acquired from gameplay videos can generalize to broader and general-purpose video understanding tasks. For example, the PhysGame-enhanced Qwen2.5-VL Bai et al. (2025) shows 1.3% performance gain (58.0% *vs.* 59.3%) on Video-MME Fu et al. (2024) benchmark. This dual transferability paradigm demonstrates that gameplay-derived training bridges both simulation-to-reality and physical-to-general understanding, offering new pathways for scalable and effective multi-modal intelligence development.

In summary, our contributions are in three-folds:

- We pioneer the use of gameplay videos as a novel training medium for physical world understanding, which overcomes the simplified settings in synthetic data while avoiding the high annotation costs of real-world video curation.
- We introduce PhysGame and GameBench, establishing a suite of scalable datasets for physics-oriented instruction tuning and evaluation, respectively.
- Experimental results validate the Game2Real and Game2General transferabilities of the proposed PhysGame dataset, bridging the simulated physics data and real-world physical understanding tasks as well as general video understanding tasks.

## 2 Related Work

**Physical World Understanding.** Recent advances in physical understanding have driven interdisciplinary research spanning psychology Bobrow (1984); Hespos et al. (2016); McCloskey et al. (1983), language reasoning Bisk et al. (2020); Forbes et al. (2019), visual reasoning Lerer et al. (2016); Yi et al. (2020), video generation Meng et al. (2024); Bansal et al. (2024), robotics Agrawal et al. (2016); Byravan et al. (2017), *etc.* The primary datasets for physical understanding Battaglia et al. (2016); Chang et al. (2016) employ symbolic representation by encoding physical knowledge through textual descriptors (*e.g.*, object attributes and interaction rules). However, such representations fundamentally lack the capacity to capture complex visual information, resulting in compromised perception. The recent datasets can be categorized into two

Table 1: **(a) Comparisons of physical understanding datasets**; **(b) Comparisons with existing gameplay video benchmarks** in terms of whether they are video-based, whether they follow an instruction-following format, and support multi-modal evaluations.

| Dataset | Photo-realistic | Scalable to Collect | Instruct Tuning |
|---|---|---|---|
| PhysBench Taesiri & Bezemer (2024) | ✓ | ✗ | ✗ |
| MMVU *et.al* Taesiri et al. (2022a) | ✓ | ✗ | ✗ |
| CLEVER Taesiri et al. (2022b) | ✗ | ✓ | ✗ |
| Comphy Chen et al. (2024e) | ✗ | ✓ | ✗ |
| **PhysGame (Ours)** | ✓ | ✓ | ✓ |

(a)

| Dataset | Video-Based | Instruct Tuning | Multi-modality |
|---|---|---|---|
| GameBunny Taesiri & Bezemer (2024) | ✗ | ✓ | ✓ |
| Taesiri *et.al* Taesiri et al. (2022a) | ✓ | ✗ | ✓ |
| GameBugDescript Taesiri et al. (2022b) | ✓ | ✓ | ✗ |
| GlitchBench Taesiri et al. (2024a) | ✗ | ✓ | ✓ |
| **PhysGame (Ours)** | ✓ | ✓ | ✓ |

(b)

principal types: 1) Real-world observational datasets capturing natural physical phenomena Chow et al. (2025); Zhao et al. (2025). This kind of dataset requires labor-intensive video curation, inherently limiting scalability in physical scenario coverage; 2) Physics-simulated datasets generated through differentiable engines Yi et al. (2020); Zheng et al. (2024); Chen et al. (2022); Bear et al. (2021). While offering controllable physical environments, they are constrained to simplified geometric primitives (*e.g.*, rigid cuboids/spheres) and homogeneous backgrounds, leading to significant sim-to-real discrepancy in visual perception. Building on this analysis, we propose a novel approach to facilitate physical world understanding by analyzing physics glitches in gameplay videos. This method establishes a scalable framework with closer alignment to real-world physical scenarios, effectively bypassing the limitations of existing datasets.

**Gameplay Video Understanding.** Digital games Hu et al. (2024a); Xu et al. (2024b) are considered pivotal in pursuing artificial general intelligence, as they act as controllable real-world simulators and create complex problem-solving contexts. Therefore, gameplay videos are typically employed as benchmarks for evaluating the capabilities of vision-language models from the perspectives of environment perception Hong et al. (2023); Akoury et al. (2023), context reasoning Liu et al. (2023); Wang et al. (2023), decision-making Chen et al. (2023); Qian et al. (2023), *etc*. The majority of existing games can be classified into two categories: 1) *Competition games* Ma et al. (2023); Shao et al. (2024); Hu et al. (2024b); Feng et al. (2024); Toshniwal et al. (2022); Li et al. (2023); Gupta (2023); Huang et al. (2024); Guo et al. (2023); Zhang et al. (2024c) in which players compete against one another, with the objective of outperforming others to achieve victory. Notable examples include StarCraft II Ma et al. (2023); Shao et al. (2024), Pokémon Battles Hu et al. (2024b), Chess Feng et al. (2024); Toshniwal et al. (2022); Li et al. (2023) and Poker Gupta (2023); Huang et al. (2024); Guo et al. (2023); Zhang et al. (2024c). Ma et al. Ma et al. (2023) introduce TextStarCraft II, a natural language-based interface that equips LLMs with the functionality to play StarCraft II, fostering more effective reasoning and decision-making capabilities; 2) *Cooperation games* Carroll et al. (2019); Gong et al. (2023); Wu et al. (2021); Puig et al. (2021); Chen et al. (2024a); Gan et al. (2021b); Puig et al. (2018); Gan et al. (2021a) are structured around collaboration, requiring players to work together to achieve shared goals. These games emphasize teamwork, communication, and joint problem-solving, where players must coordinate efforts to succeed and reach mutual accomplishments. In Overcooked-AI Carroll et al. (2019), the preparation of an onion soup requires two agents to collaborate by loading three onions into a cooking pot, thereby initiating a cooking process that spans 20-time steps.

The most relevant line of research to ours is on the topic of game bug detection (Taesiri et al., 2022a; 2024a; 2022b; Taesiri & Bezemer, 2024; Taesiri et al., 2024b). Existing methods, however, either focus the classification/retrieval tasks Taesiri et al. (2022a) or limited in the static image domain Taesiri & Bezemer (2024); Taesiri et al. (2024a) (*cf*. Table 1b). The prior work GameBugDescriptions Taesiri et al. (2022b) employs LLMs to detect bugs in game videos with the reliance on pre-extracted event-wise *textual* descriptions and lacks support for multi-modal evaluations. In contrast, our PhysGame addresses all these limitations and supports multi-modal instruction tuning in videos.

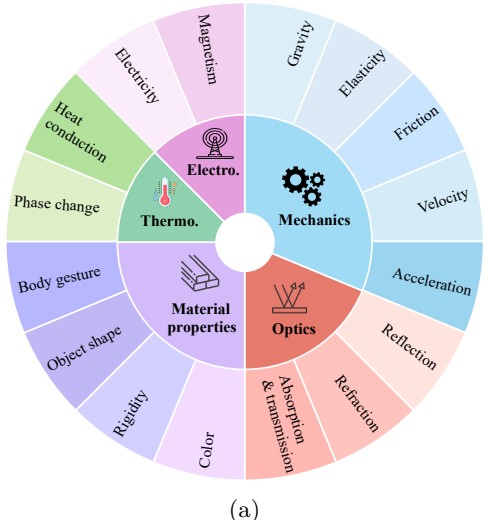

| Statistics of PhysGame | Value |
|---|---|
| Total number of QA pairs | 140,057 |
| Unique Videos | 38,957 |
| Question Length (avg/max) | 9.81 / 113 |
| Answer Length (avg/max) | 53.49 / 143 |
| Video Length (Seconds, avg/max) | 23.33 / 2611.17 |
| **Statistics of GameBench** | **Value** |
| Total number of QA pairs | 880 |
| Unique Videos | 880 |
| Question Length (avg/max) | 9.06 / 17 |
| Answer Length (avg/max) | 14.46 / 38 |
| Video Length (Seconds, avg/max) | 25.82 / 239.57 |

(a)                                              (b)

Figure 3: **(a) The taxonomy** and **(b) key statistics** of the PhysGame and GameBench dataset.

## 3 Dataset Construction

### 3.1 PhysGame

Building on the intuitive cognition of physical understanding, PhysGame introduces a comprehensive taxonomy for task categorization including five primary domains, *i.e.*, mechanics, optics, material properties, thermodynamics, and electromagnetism, and sixteen fine-grained categories (*cf*. Figure 3a).

- *Mechanics*: This category deals with forces and torques as well as their effects on motion, which provides the foundational principles to interpret and analyze the motion of objects in videos. Typical cases include gravity, elasticity, friction, velocity and acceleration over time.
- *Optics*: It focuses on the behavior and properties of light as well as its interactions with matter. It includes reflection, refraction, and absorption & transmission.
- *Material properties*: It refers to the inherent material characteristics including color, rigidity, object shape, and human body gesture.
- *Thermodynamics*: This category involves the study of heat, temperature, and energy transfer between systems. Typical cases include heat conduction and phase change.
- *Electromagnetism*: It encompasses phenomena related to the electric and magnetic fields and their interactions with matter.

**Video Collection.** The videos of PhysGame dataset are curated from the `GamePhysics` subreddit[1], a community where users share gameplay videos exhibiting various unusual events and glitches across different games, accompanied by the users' discussions (meta-information). This platform offers abundant glitch-containing gameplay footage alongside meta-information that broadly describes glitch characteristics, enabling the scalability of our proposed PhysGame dataset.

**Instruction-following Generation.** We follow the self-instruction paradigm Li et al. (2024a) to construct PhysGame by prompting GPT-4o OpenAI (2024). In terms of the instruction generation, we aspire for the questions to be as diverse as possible. We set up three question types, varying from direct to indirect assessment of the glitches in videos: i) explicitly inquiring about glitches in the video, ii) probing anomalies present in the video, iii) or merely straightforward questions regarding the video content. We provide three examples corresponding to the mentioned three scenarios: i) *What is the description of the glitch observed*

---

[1] `www.reddit.com/r/GamePhysics/`

| | |
|---|---|
| **Question**: What is the description of the glitch observed in the video? | *Glitch content: No realistic wave propagation* |
| **Title**: | [GTA IV] I hope they bring back the water physics. |
| **Answer w/ title** | In the video, when the player pilots a helicopter to land on the water surface, the spinning rotor blades create a realistic disturbance, pushing the water away and forming a clear circular pattern beneath the aircraft. ...... At specific moments, the surface of the water lacks believable wave propagation, especially when interacting with nearby objects or terrain. These inconsistencies slightly detract from the realism ... |
| **Answer w/o title** | The glitch involves a helicopter interacting with water in an unrealistic and physically implausible way. After descending and making contact with the water's surface, instead of behaving like a damaged or crashing aircraft—sinking, breaking apart, or stopping—it begins to bounce erratically and spin uncontrollably. The helicopter skims and ricochets across the water as if it were a rubber ball, completely ignoring expected aerodynamic drag or water resistance. ... |

Figure 4: **The generated responses in PhysGame dataset** with or without meta-information hints (*i.e.*, titles). The title indicates a glitch related to *water physics*. With the title hints, the generated responses accurately identify that the surface of the water lacks realistic wave propagation; in contrast, the response contains hallucinated content without the title hints.

Table 2: **Quality validation** for the proposed PhysGame dataset.

| Meta Data Acc | QA Acc *w/* Meta | QA Acc *w/o* Meta |
|:---:|:---:|:---:|
| 86% | 91% | 64% |

*in the video?* ii) *Are there any abnormalities present in the videos?* iii) *Please provide a description of the video content.*

As for the response generation, the preliminary experiments suggest that the intuitive prompting method leads to remarkable errors. To alleviate this, we propose a *meta-information guided prompting* strategy. Specifically, we found that the meta-information (*e.g.*, title) associated with each video offers insight into the fundamental content. Therefore, we propose to incorporate video-wise meta-information in the prompt, resulting in more accurate instruction-following generation. As shown in Figure 4, the meta-information helps GPT-4o to detect the "*water physics*" glitch, while its absence leads to the degradation of physical commonsense understanding. The key statistics of the constructed PhysGame are presented in Table 3b.

**Quality Control.** Given the scale of our instruction dataset, per-sample manual verification proves infeasible. We implement a systematic quality validation framework by sampling 2,000 instances and conducting two-tier manual verification: 1) metadata relevance to video semantics, and 2) QA alignment with observed physical anomalies. The experiments in Table 2 show 86% metadata accuracy and 91% QA accuracy, demonstrating the overall reliability of our PhysGame. In addition, the comparison between QA accuracy with and without metadata (91% *vs.* 64%) confirms that incorporating meta-information improves the quality of generated instruction data.

### 3.2 GameBench

**Video Collection and Filter.** The video collection procedure of GameBench follows the same process as that in PhysGame. To prevent data leakage, we diligently exclude any videos included in PhysGame. We conduct manual checks based on the following two criteria: 1) *Duplicate check*: The Reddit[1] discussion forum may feature multiple references to the same video, resulting in duplicate downloading. We manually check to confirm that each video in GameBench is distinct; 2) *Content check*: The pool of downloaded videos may incorporate non-game elements, which we rigorously filter out of our GameBench benchmark.

**Annotation Scheme.** Based on the collected gameplay videos, we create the question-answer pairs in a four-way multiple-choice format to facilitate convenient evaluation. Specifically, the correct options describe the video-specific glitches that contravene physical commonsense principles. It is important to note that some videos may exhibit multiple glitches. We therefore instruct expert annotators to review the entire video to ensure all the appearing glitches are included in the correct answer.

To enhance the plausibility of the distractor options, we have provided expert annotators with three guiding principles: **1)** Instead of imagining arbitrary glitches, the glitch in the distractor options should be highly correlated to the individuals or actions observed in the videos. For example in Figure 2 (right), the distractor option B includes "*car*" and "*shadow*" that are genuinely present in the video. This annotation principle forces MLLMs to comprehend the glitchy content, rather than merely selecting answers by identifying contained objects or actions; **2)** The four choice options should be of similar length, which helps prevent any preference biases in MLLMs; **3)** To mitigate choice bias, the distribution of the correct option among the four choices should be equitable, (*i.e.*, 25% likelihood for each option).

**Quality Control.** To guarantee the quality of our dataset, we conduct a two-fold quality control process: **1)** *LLM-assisted inspection*: We exclude question-answering pairs that can be correctly answered by GPT-4o OpenAI (2024) solely based on the question and options without the need to view the video. By statistics, we limit the accuracy of GPT-4o in the question-only scenario to less than 25%; **2)** *Human inspection*: All the initially annotated question-answering pairs undergo rigorous cross-inspection by different human annotators. For the correct options, the inspectors must assess whether they comprehensively and accurately describe all instances of physical commonsense violations present. For the distractor options, the inspectors are required to evaluate whether they are sufficiently deceptive, specifically by including objects or actions depicted in the video. Through the rigorous construction and review process, we present the GameBench benchmark which is of high quality and well-balanced. The specific statistics are available in Table 3b.

## 4 Experiments

### 4.1 Experimental Settings

**Baseline and Implementation Details.** To evaluate the efficiency of the PhysGame dataset, we performed supervised fine-tuning on three MLLMs: Qwen2-VL-7B Wang et al. (2024a), Qwen2.5-VL-7B Bai et al. (2025), and InternVL2.5-8B Chen et al. (2024c). To prevent the model from becoming overly biased toward the gameplay video domain during fine-tuning, we additionally incorporated 20K samples from the general video instruction tuning dataset LLaVA-Hound Zhang et al. (2024b), forming a total training set of 160K samples. The Qwen series models Wang et al. (2024a); Bai et al. (2025) were trained for one epoch with a batch size of 128 and a learning rate of 2e-7, whereas InternVL2.5-8B Chen et al. (2024c) employed a higher learning rate of 1e-6 with an equivalent batch size. All models processed 32-frame video sequences as input. Experiments were conducted on 8 NVIDIA A100 GPUs.

**Evaluation Benchmarks.** To demonstrate the broad applicability of our PhysGame dataset, we conducte experiments across both physical world understanding and general video understanding tasks:

- *Physical World Understanding*: Depending on the video data source, we evaluate on two categories of benchmarks: gameplay-based dataset (*i.e.*, our self-constructed GameBench) and real-world datasets (*i.e.*, PhysBench Chow et al. (2025) and MMVU Zhao et al. (2025)).
- *General Video Understanding*: We further validate the broad value of PhysGame through evaluations on established general video understanding benchmarks including MVBench Li et al. (2024b), VideoMME Fu et al. (2024) and LongVideoBench Wu et al. (2024).

### 4.2 Physical Understanding

**Evaluation Results on GameBench.** For comparisons, we benchmark the proposed GameBench on both proprietary MLLMs (*i.e.*, Claude3.5-Sonnet Anthropic (2024), Claude3.5-SonnetV2 Anthropic (2024), Gemini-1.5-pro Team et al. (2024), Gemini-1.5-pro-flash Team et al. (2024), GPT-4V Achiam et al. (2023), GPT-4o-0806 OpenAI (2024), GPT-4o-mini-0718 OpenAI (2024) and Qwen-VL-max Bai et al. (2023)) and

Table 3: **Evaluation (%) results on the proposed GameBench benchmark** including the domains of mechanics, optics, material properties, thermodynamics, and electromagnetism.

| Models | AVG | Mechanics | Optics | Material | Thermo | Electro |
|---|---|---|---|---|---|---|
| Claude3.5-Sonnet Anthropic (2024) | 54.3 | 53.7 | 49.6 | 54.5 | 71.4 | 55.3 |
| Claude3.5-SonnetV2 Anthropic (2024) | 47.6 | 47.0 | 42.1 | 51.9 | 53.1 | 46.8 |
| Gemini-1.5-pro Team et al. (2024) | 55.2 | 55.0 | 49.6 | 58.7 | 63.3 | 51.1 |
| Gemini-1.5-pro-flash Team et al. (2024) | 48.5 | 50.4 | 41.4 | 50.3 | 49.0 | 42.6 |
| GPT-4V Achiam et al. (2023) | 45.9 | 45.9 | 43.6 | 43.9 | 53.1 | 53.2 |
| GPT-4o OpenAI (2024) | 56.1 | 55.4 | 48.9 | 60.3 | 67.3 | 55.3 |
| GPT-4o-mini OpenAI (2024) | 40.3 | 40.7 | 39.1 | 41.3 | 36.7 | 40.4 |
| Qwen-VL-Max | 50.9 | 46.8 | 52.6 | 55.0 | 69.4 | 51.1 |
| LLaVA-Next-Video Liu et al. (2024b) | 32.2 | 31.8 | 25.6 | 37.0 | 42.9 | 23.4 |
| Video-LLaVA Lin et al. (2023) | 29.0 | 29.2 | 28.6 | 31.2 | 30.6 | 17.0 |
| LLaVA-OneVision | 47.7 | 46.3 | 48.1 | 49.2 | 59.2 | 42.6 |
| InternVL2 Chen et al. (2024c) | 33.4 | 34.2 | 37.6 | 29.1 | 30.6 | 34.0 |
| VideoChat2 | 34.3 | 33.1 | 31.6 | 37.0 | 44.9 | 31.9 |
| ST-LLM | 32.8 | 30.3 | 30.8 | 38.6 | 49.0 | 23.4 |
| Chat-UniVi Jin et al. (2024) | 29.5 | 30.7 | 30.1 | 30.7 | 20.4 | 21.3 |
| PPLLaVA | 38.4 | 37.7 | 38.3 | 36.0 | 51.0 | 42.6 |
| Qwen2-VL-7B Wang et al. (2024a) | 37.5 | 35.5 | 34.6 | 37.0 | 34.7 | 34.0 |
| *w/* PhysGame | **43.8** | **45.5** | **47.4** | **48.7** | **40.8** | **36.2** |
| Δ | +6.3 | +10.0 | +12.8 | +11.7 | +6.1 | +2.2 |
| Qwen2.5-VL-7B Bai et al. (2025) | 44.4 | 46.5 | 40.6 | 46.0 | 34.7 | 38.3 |
| *w/* PhysGame | **48.1** | **51.5** | **42.1** | **48.1** | **34.7** | **44.7** |
| Δ | +3.7 | +5.0 | +1.5 | +2.1 | 0.0 | +6.4 |
| InternVL2.5-8B Chen et al. (2024c) | 38.6 | 35.9 | 33.1 | 43.4 | 59.2 | 40.4 |
| *w/* PhysGame | **48.0** | **44.4** | **43.6** | **56.6** | **67.3** | **40.4** |
| Δ | +9.4 | +8.5 | +10.5 | +13.2 | +8.1 | +0.0 |

open-source models including LLaVA-Next-Video Liu et al. (2024b), Video-LLaVA Lin et al. (2023), LLaVA-OneVision Li et al. (2024a), InternVL2 Chen et al. (2024d), VideoChat2 Li et al. (2024b), ST-LLM Liu et al. (2024c), Chat-UniVi Jin et al. (2024) and PPLLaVALiu et al. (2024d). We follow Video-MME Fu et al. (2024) to utilize the official frame configurations provided for each MLLM. We employ the selection accuracy as the evaluation metric for our curated multi-choice questions.

The evaluation results on the GameBench benchmark are presented in Table 3. Key observations include: 1) *Current methods struggle on GameBench*: Among proprietary models, GPT-4o OpenAI (2024) and Gemini-1.5-pro Team et al. (2024) achieve the best performance with average accuracy scores of only 56.1% and 55.2%, respectively. As a reminder, the GPT-4o score of 56.1% is achieved on GameBench after filtering out questions that were found to be answerable by text alone (using GPT-4o itself during quality control), which further underscores the challenging nature of the benchmark. This indicates persistent challenges in understanding physical commonsense within complex, dynamically evolving scenarios; 2) *Instruction tuning on PhysGame benefits*: Baseline models (Qwen2-VL-7B Wang et al. (2024a), Qwen2.5-VL-7B Bai et al. (2025), InternVL2.5-8B Chen et al. (2024c)) show substantial improvements after PhysGame-based instruction tuning. For instance, the PhysGame-tuned Qwen2-VL achieves an 6.3% higher average accuracy on GameBench compared to its baseline version, demonstrating the dataset's value for gameplay-based physical reasoning; 3) *Inconsistent improvements*: An interesting observation is that InternVL2.5-8B exhibits the most significant improvement in average accuracy after fine-tuning on PhysGame, achieving a gain of 9.4%, with consistent performance increases across all subcategories. In contrast, the Qwen-series models Wang et al. (2024a); Bai et al. (2025) show relatively limited improvement, likely because their pretrained

Table 4: **Evaluation (%) of Game2Real transferability** on PhysBench and MMVU. "*w/* PhysGame" denotes finetuning baseline models on the proposed PhysGame dataset.

| Models | PhysBench | | | | | MMVU |
| --- | --- | --- | --- | --- | --- | --- |
| | Average | Property | Relationships | Scene | Dynamics | Val |
| GPT-4V Achiam et al. (2023) | 41.3 | 49.6 | 45.8 | 26.3 | 42.2 | – |
| GPT-4o OpenAI (2024) | 49.5 | 56.9 | 64.8 | 30.2 | 47.0 | 67.4 |
| GPT-4o-mini OpenAI (2024) | 43.2 | 53.5 | 44.2 | 30.6 | 42.9 | 61.6 |
| Gemini-1.5-flash Team et al. (2024) | 46.1 | 57.4 | 52.2 | 34.3 | 40.9 | 58.8 |
| Gemini-1.5-pro Team et al. (2024) | 49.1 | 57.3 | 63.6 | 36.5 | 41.6 | 65.4 |
| Llava-Next-Video Liu et al. (2024b) | 35.4 | 38.3 | 30.8 | 34.0 | 37.2 | 28.6 |
| Phi-3V Abdin et al. (2024) | 38.4 | 43.7 | 37.9 | 34.9 | 36.9 | – |
| VILA-1.5-8B Lin et al. (2024) | 32.9 | 33.4 | 29.9 | 30.9 | 35.9 | – |
| VILA-1.5-13B Lin et al. (2024) | 37.2 | 40.5 | 40.2 | 32.0 | 36.1 | – |
| Video-LLaVA Lin et al. (2023) | 37.0 | 36.8 | 36.1 | 33.7 | 40.5 | – |
| Chat-Univi-7B Jin et al. (2024) | 22.2 | 19.3 | 21.0 | 18.9 | 28.5 | – |
| Chat-Univi-13B Jin et al. (2024) | 10.4 | 4.3 | 11.5 | 15.7 | 11.5 | – |
| Claude-3.5-sonnet Anthropic (2024) | 38.1 | 46.5 | 41.1 | 27.9 | 37.6 | 65.2 |
| InternVL-Chat1.5-26B Chen et al. (2024d) | 47.5 | 53.1 | 70.1 | 37.0 | 44.8 | – |
| Qwen2-VL-7B Wang et al. (2024a) | 46.6 | 54.3 | 60.4 | 39.4 | 47.0 | 42.1 |
| *w/* PhysGame | **49.1** | **57.5** | **63.2** | **41.8** | **49.1** | **48.6** |
| $\Delta$ (*vs.* Qwen-2VL-7B) | +2.5 | +3.2 | +2.8 | +2.4 | +2.1 | +6.5 |
| Qwen2.5-VL-7B Bai et al. (2025) | 45.9 | 54.6 | 63.2 | 37.0 | 46.9 | 48.8 |
| *w/* PhysGame | **47.6** | **57.7** | **64.1** | **38.3** | **48.2** | **50.5** |
| $\Delta$ (*vs.* Qwen-2.5-VL-7B) | +1.7 | +3.1 | +0.9 | +1.3 | +1.3 | +1.7 |
| InternVL2.5-8B Chen et al. (2024c) | 43.9 | 55.9 | 48.7 | 29.4 | 41.2 | 41.1 |
| *w/* PhysGame | **46.0** | **57.9** | **66.6** | **30.7** | **51.2** | **49.6** |
| $\Delta$ (*vs.* InternVL2.5-8B) | +2.1 | +2.0 | +17.9 | +1.3 | +10.0 | +8.5 |

vision-language alignment and reasoning capabilities are already highly optimized through large-scale multi-modal instruction tuning, leaving limited room for further enhancement from task-specific physical reasoning data.

**Game2Real Transferability.** In addition to gameplay and simulator datasets, our PhysGame dataset demonstrates the Game2Real transferability on real-world physical understanding benchmarks including PhysBench Chow et al. (2025) and MMVU Zhao et al. (2025). PhysBench is proposed to evaluate MLLMs' physical world understanding capabilities, which integrates QAs regarding property, relationships, scene, and dynamics. MMVU assesses knowledge-intensive reasoning on specialized-domain videos. We conduct experiments on its validation set.

The Game2Real transfer performance is shown in Table 4. Across four subcategories of PhysBench and MMVU, fine-tuning models on PhysGame (denoted by "*w/* PhysGame" in Table 4) consistently improves their performance over the baselines, demonstrating the effectiveness of our proposed PhysGame in enhancing physical understanding and generalization. For example, Qwen2-VL-7B and InternVL2.5-8B see significant absolute gains of +2.5% and +2.1% respectively on average after fine-tuning, while Qwen2.5-VL-7B achieves the minimal improvement of +1.7%. These results confirm the strong Game2Real transferability facilitated by the PhysGame dataset.

### 4.3 General Video Understanding Results

We investigate the transferability of physics knowledge learned from PhysGame to general video understanding (Game2General transferability). Our experiments span MV-Bench Li et al. (2024b), Video-MME Fu et al. (2024), and LongVideoBench Wu et al. (2024), where MV-Bench focuses on multifaceted video un-

Table 5: **Evaluation (%) of Game2General transferability** on MVBench, Video-MME and LongVideoBench. "w/o subs" denotes results without subtitles. "w/ PhysGame" denotes finetuning baseline models on the proposed PhysGame dataset.

| Models | MVBench | Video-MME (*w/o* subs) | | | | LongVideoBench |
|---|---|---|---|---|---|---|
| | | AVG | Short | Medium | Long | Val |
| GPT-4V Achiam et al. (2023) | 43.5 | 59.9 | 70.5 | 55.8 | 53.5 | – |
| Video-LLaVA Lin et al. (2023) | – | 39.9 | 45.3 | 38.0 | 36.2 | 39.1 |
| LLaVA-1.5 Liu et al. (2024a) | 36.0 | – | – | – | – | 40.3 |
| PLLaVA Xu et al. (2024a) | 46.6 | – | – | – | – | 40.2 |
| Qwen-VL-Max Bai et al. (2023) | – | 51.3 | 55.8 | 49.2 | 48.9 | – |
| ShareGPT4Video-8B Chen et al. (2024b) | 51.2 | 39.9 | 48.3 | 36.3 | 35.0 | 39.7 |
| InternVL-Chat-V1.5 Chen et al. (2024d) | – | 45.6 | 50.7 | 60.2 | 46.4 | 51.2 |
| VideoChat2 Li et al. (2024b) | 51.1 | 39.5 | 48.3 | 37.0 | 33.2 | 39.3 |
| LongLLaVA-9B Wang et al. (2024b) | 49.1 | 43.7 | 52.4 | 42.2 | 36.4 | – |
| Video-CCAM-9B Fei et al. (2024) | 64.6 | 50.3 | 61.9 | 49.2 | 39.6 | – |
| NVILA Liu et al. (2024e) | 68.1 | 64.2 | 75.7 | 62.2 | 54.8 | 57.7 |
| Apollo Zohar et al. (2024) | – | 61.3 | – | – | – | 58.5 |
| LongVA-7B Zhang et al. (2024a) | – | 52.6 | 61.1 | 50.4 | 46.2 | 51.3 |
| LLaVA-Video-7B Zhang et al. (2024d) | 58.6 | 63.3 | – | – | – | 58.2 |
| Qwen2-VL-7B Wang et al. (2024a) | 64.5 | 58.0 | 70.2 | 55.8 | 48.1 | 55.3 |
| *w/* PhysGame | **66.4** | **59.3** | **70.7** | **57.6** | **49.7** | **56.1** |
| Δ (*vs.* Qwen-2VL-7B Wang et al. (2024a)) | +1.9 | +1.3 | +0.5 | +1.8 | +1.6 | +0.8 |
| Qwen2.5-VL-7B Bai et al. (2025) | 66.0 | 64.7 | 76.2 | 63.3 | 54.6 | 56.0 |
| *w/* PhysGame | **67.3** | **65.8** | **76.6** | **65.7** | **55.2** | **58.3** |
| Δ (*vs.* Qwen-2.5-VL-7B Bai et al. (2025)) | +1.3 | +1.1 | +0.4 | +2.4 | +0.6 | +2.3 |
| InternVL2.5-8B Chen et al. (2024c) | 72.0 | 64.0 | 75.8 | **62.8** | **53.4** | 57.2 |
| *w/* PhysGame | **72.8** | **64.2** | **76.8** | 62.6 | 53.3 | **59.5** |
| Δ (*vs.* InternVL2.5-8B Chen et al. (2024c)) | +0.8 | +0.2 | +1.0 | -0.2 | -0.1 | +2.3 |

derstanding capabilities in short videos while Video-MME and LongVideoBench emphasize long-form video comprehension. All evaluations follow the official protocols.

As shown in Table 5, finetuning models on PhysGame consistently boosts their performance across all benchmarks. For example, Qwen-2-VL-7B achieves a 1.9% improvement on MVBench and sees gains across all three temporal scopes of Video-MME, with a notable 1.8% gain on medium-length videos. Qwen-2.5-VL-7B also benefits from PhysGame, attaining a 2.3% accuracy boost in LongVideoBench. Similarly, InternVL2.5-8B improves from 72.0% to 72.8% on MVBench, marking a new state-of-the-art result. We also observe a slight performance drop on the medium and long video splits of the Video-MME dataset after fine-tuning on PhysGame, which may be attributed to the fact that most videos in PhysGame are short, leading InternVL2.5-8B to develop a bias toward short-duration videos. Overall, these findings confirm that physics-oriented pretraining via PhysGame not only improves physical reasoning but also enhances the generalization of video-language models to diverse and longer-duration video tasks.

## 4.4 Ablation Studies

**Data-scaling effects.** To quantify the data-scaling behavior of our PhysGame dataset, we conduct additional experiments using Qwen2.5-VL-7B fine-tuned with different amounts of PhysGame data (0K, 40K, 80K, 140K) to analyze the scaling trends. As noted in Section 4.1, for real-world and general-video tasks (PhysBench, MMVU, MVBench, Video-MME, LongVideoBench), we mix PhysGame with 20K LLaVA-Hound samples to maintain domain balance. The results in Figure 5 demonstrate that the performance consistently improves as more PhysGame data is used. The gains are monotonic and do not plateau even at

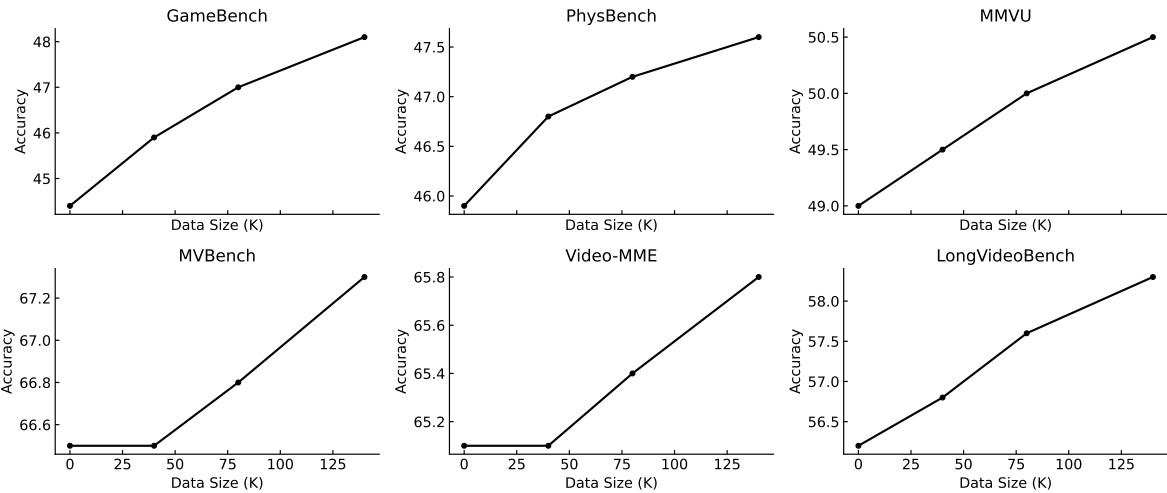

Figure 5: **Data-scaling effects** of our PhysGame dataset.

Table 6: **Ablation study** on meta-data and training data mixture based on Qwen2.5-VL.

| Method | PhysBench | MMVU | MVBench | Video-MME | LongVideoBench |
|---|---|---|---|---|---|
| Baseline | 45.9 | 48.8 | 66.0 | 64.7 | 56.0 |
| Full Mode | **47.6** | **50.5** | **67.3** | **65.8** | **58.3** |
| w/o meta-data | 42.9 | 46.7 | 65.2 | 63.5 | 55.2 |
| w/o data mixture | 45.1 | 49.2 | 66.7 | 65.1 | 57.4 |

140K samples, indicating that the dataset's scaling potential has not been saturated. This trend validates the scalability of PhysGame and suggests that further expansion could yield additional improvements.

**Ablations on dataset mixture.** In Section 4.1, we constructed the overall training set by combining PhysGame-140K with 20K data from LLaVA-Hound Zhang et al. (2024b). We ablate on this dataset mixture strategy by fine-tuning using only PhysGame (140K) dataset without any LLaVA-Hound data. The experimental results in Table 6, the data mixing strategy plays a clear role, particularly in enhancing generalization to real-world benchmarks (*e.g.*, +2.5% on PhysBench compared to using only PhysGame).

**Ablations on training dataset composition.** We conduct fixed-budget and seed-controlled resampling to ablate on the training dataset configuration. Specifically, we fix the total training budget to 160K samples and systematically vary the mixing ratio between PhysGame (PG) and LLaVA-Hound (LH) as [0K/160K, 40K/120K, 80K/80K, 140K/20K]. A fixed random seed ensures the same LH subset is used across runs, eliminating sampling randomness.

As the results in Table 7 indicate, using only LLaVA-Hound does not yield performance gains. For instance, with Qwen2.5-VL, only using the general video instruction tuning dataset LLaVA-Hound maintains the same results as the zero-shot evaluation (45.9%). Using only PhysGame leads to a performance drop (45.1% vs. 45.9%), likely because post-training solely on PhysGame induces a bias towards gameplay videos. Consequently, we selected a mixing ratio of 140K samples from PhysGame and 20K samples from LLaVA-Hound to achieve optimal performance. One interesting finding is that the performance of the pure PhysGame configuration (140K/0K) is slightly lower than that of the pure general-data configuration (0K/160K). This may be primarily due to two factors. First, the difference in total training samples (140K vs. 160K) gives the general-data setting a scale advantage in learning diverse visual representations. Second, training solely on PhysGame, which is visually centered on gameplay scenes, can induce a mild domain bias, slightly hindering generalization to the broader visual distributions found in real-world benchmarks like PhysBench.

Table 7: **Ablation study** on the configuration of PhysGame (PG) and LLaVA-Hound (LH) datasets.

| Configuration of PG/LH | 0K/160K | 40K/120K | 80K/80K | 140K/20K | 140K/0K |
|---|---|---|---|---|---|
| **AVG on PhysBench** | 45.9% | 46.9% | 47.3% | **47.6%** | 45.1% |

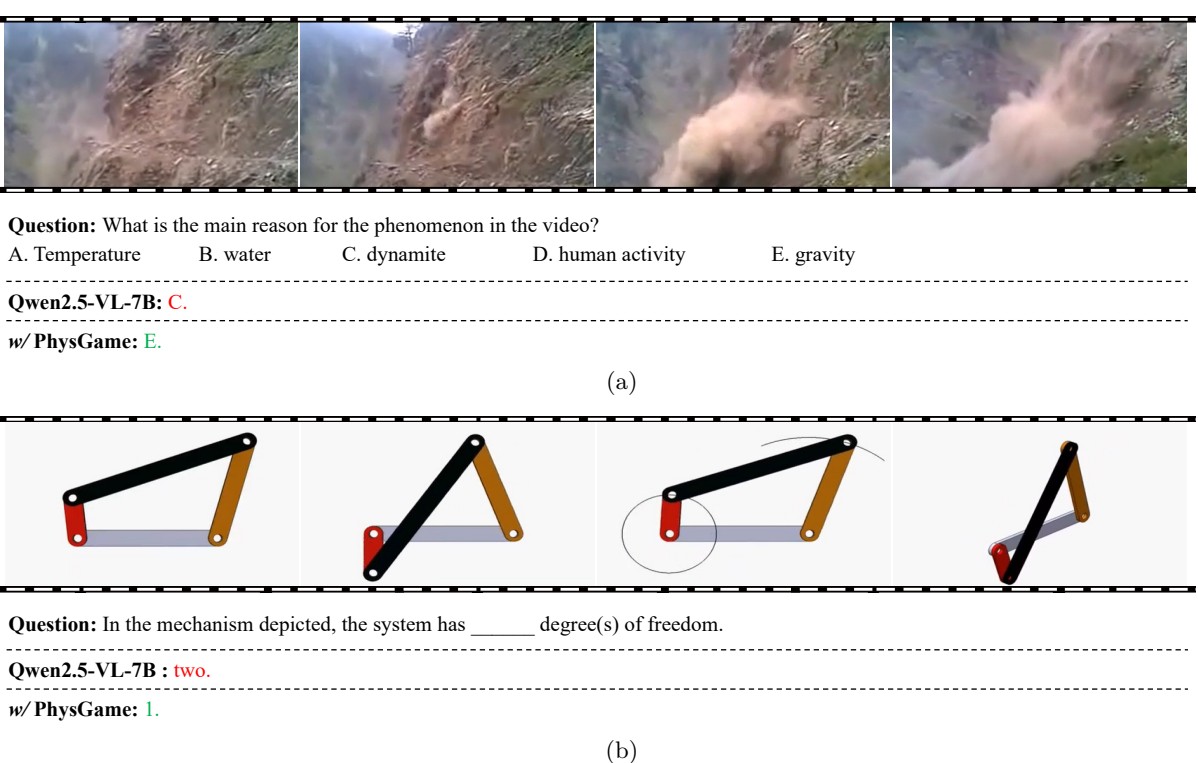

Figure 6: Comparisons between the baseline Qwen2.5-VL-7B and its PhysGame fine-tuned counterpart on MMVU benchmark.

**Ablations of the meta-information-guided prompting.** In Section 3.1, we conduct a manual quality validation to demonstrate that meta-information enhances the quality of generated QA pairs. Here we present quantitative experiments using Qwen2.5-VL trained on PhysGame with/without meta-information guidance. Our experimental results in Table 6 demonstrate that integrating meta-information leads to a 4.7% absolute improvement (42.9% *vs.* 47.6%) on PhysBench, which further highlights the necessity of using meta-information guidance during the prompt generation process.

**Visualizations.** Figure 6 presents qualitative comparisons between the baseline Qwen2.5-VL-7B and its PhysGame fine-tuned counterpart on representative cases from the MMVU benchmark. In Figure 6a, the baseline model incorrectly attributes the observed landslide phenomenon to dynamite, indicating a bias toward surface-level visual cues (*e.g.*, explosion-like motion and dust dispersion). In contrast, the PhysGame-enhanced model correctly identifies gravity as the underlying cause, demonstrating an improved ability to infer causal relationships grounded in physical principles. In Figure 6b, where the task involves determining the degree of freedom of a linkage mechanism, the baseline model outputs two, failing to account for the kinematic constraints imposed by joint connections. The PhysGame model, however, accurately infers one, reflecting a stronger grasp of motion constraints and mechanical structure reasoning. Together, these examples illustrate that PhysGame effectively bridges the gap between visual perception and intuitive physics reasoning, enabling the model to generate more physically consistent interpretations of both natural and synthetic scenes.

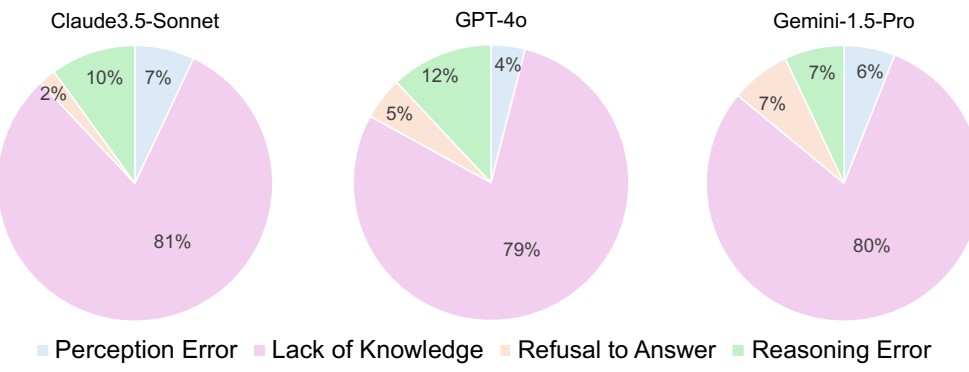

Figure 7: Error type analysis.

**Error type analysis.** To investigate the bottlenecks of current MLLMs in physical world understanding, we analyze the top-performing models on the proposed GameBench benchmark: Gemini 1.5 Pro Team et al. (2024), GPT-4o OpenAI (2024), and Claude 3.5 Sonnet Anthropic (2024). We manually examine the error distribution across 100 selected cases, categorizing errors into four types:

- *Perception error*: MLLMs fail to correctly recognize entities, characters, or actions in the video.
- *Lack of knowledge*: MLLMs correctly perceive the video content but make incorrect judgments about the physical glitch.
- *Reasoning error*: MLLMs identifies both the video content and the glitch but fails to detect reasoning flaws in the provided answer options (*e.g.*, causal errors).
- *Refusal to answer*: MLLMs declines to respond due to limited capabilities.

Figure 7 presents a comparative analysis of error distributions across three leading MLLMs including Claude 3.5-Sonnet, GPT-4o, and Gemini 1.5-Pro. Across all models, the dominant error type is "Lack of Knowledge", accounting for approximately 79–81% of total errors. This trend indicates that factual incompleteness or outdated knowledge remains the principal bottleneck in current large-scale multimodal reasoning systems, rather than perception or reasoning capabilities. Among the secondary categories, Reasoning Errors contribute between 7–12%, with GPT-4o showing the highest proportion (12%). Perception Errors are relatively minor (4–7%) across all models, reflecting the general maturity of visual grounding components. Refusal to Answer occurs least frequently, with Claude 3.5-Sonnet exhibiting the most conservative behavior (2%), while GPT-4o and Gemini 1.5-Pro show slightly higher refusal rates (5–7%), possibly due to stricter safety or uncertainty calibration mechanisms. Overall, the results highlight a consistent pattern: while perception and reasoning are gradually improving, the knowledge base's coverage and retrieval precision remain the dominant limitations shaping multimodal model reliability.

## 5 Conclusion

In this work, we propose PhysGame, a scalable and photo-realistic instruction-tuning dataset constructed from gameplay videos to advance physical world understanding of MLLMs. By leveraging glitch-induced annotations, PhysGame enables systematic discovery of physical inconsistencies across diverse scenarios involving mechanics, optics, material properties, thermodynamics, and electromagnetism. Compared to existing real-world or synthetic datasets, PhysGame achieves a better trade-off between realism and scalability. Extensive experiments demonstrate that PhysGame not only improves real-world physical reasoning (Game2Real), but also enhances general-purpose video understanding (Game2General). We position PhysGame as a scalable and effective paradigm for equipping MLLMs with robust physical reasoning and generalizable understanding abilities.

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
