# −Supplementary Material−
# Order from Chaos: Physical World Understanding from Glitchy Gameplay Videos

**Meng Cao[1]\***, **Haoran Tang[2]\***, **Haoze Zhao[1]\***, **Mingfei Han[1]**, **Ruyang Liu[2]**, **Qiang Sun[1,3]†**,
**Xiaojun Chang[1]**, **Ian Reid[1]**, **Xiaodan Liang[1,5]†**
**[1]Mohamed bin Zayed University of Artificial Intelligence  [2]Peking University**
**[3]University of Waterloo  [4]University of Toronto  [5]Sun Yat-sen University**
**\*Authors contributed equally to this research.  †Corresponding author.**

Reviewed on OpenReview: **https://openreview.net/forum?id=Oe5TdpPv1b**

**License discussion.** We understand and appreciate the importance of responsible data use. To clarify, all videos used in our dataset are publicly available and shared by users on Reddit under subreddit communities that operate under Reddit's *User Agreement* and *Content Policy*. According to Reddit's terms [1], users grant Reddit **a broad license to host and redistribute uploaded content**, and its API **allows lawful academic access to public posts for research purposes**.

To better align with ethical standards and community expectations, we make the following modifications to address potential copyright concerns:

- **Switch to link-only indexing:** We update our data release to only index and compile links to the original publicly available videos and metadata on Reddit, without hosting or redistributing any video content ourselves.
- **Add opt-out mechanism and usage documentation:** We will additionally update our dataset release with an *opt-out* mechanism, allowing content owners to request removal of their video references, and provide clear documentation of the dataset's terms of use. The specific contents are as follows: We uphold the rights of individuals and copyright holders. If you are featured in any of our video annotations or hold copyright to a video and wish to have its annotation removed from our dataset, please reach out to us. We commit to reviewing your request promptly and taking suitable action.

**Broader Impact statement.** This work uses publicly available gameplay videos that contain only synthetic and non-personal virtual environments to construct scalable datasets for studying physical reasoning in MLLMs. Since the data do not depict real individuals or communities, the dataset does not encode demographic or cultural biases, and we provide genre-level statistics for transparency. All content is accessed under Reddit's standard usage terms, and we release only metadata and source links rather than re-distributing the videos themselves. The method does not generate or manipulate real imagery and is not intended for use in high-risk domains such as autonomous driving or robotics, so misuse risks such as harmful content or deepfakes are minimal. Overall, this work aims to advance research on physical commonsense while maintaining ethical data governance and transparent reporting.

**Experimental results based on additional baselines.** In the main paper, we already evaluate Phys-Game on three representative open-source MLLMs (Qwen2-VL-7B Wang et al. (2024), Qwen2.5-VL-7B Bai et al. (2025), InternVL2.5-8B Chen et al. (2024)), demonstrating consistent improvements on both Game2Real and Game2General transfer tasks. To further strengthen the claim that our approach is robust across architectures and model scales, we additionally experiment on four more models of different families and parameter sizes: Qwen2.5-VL-3B Bai et al. (2025), InternVL2.5-4B Chen et al. (2024), LLaVA-Video-7B Zhang et al. (2024), and LLaVA-OneVision-1.5-8B An et al. (2025). The results with and without Phys-Game finetuning on GameBench, PhysBench, MMVU, MVBench, Video-MME, and LongVideoBench are summarized in Table 3.

---

[1]https://support.reddithelp.com/hc/en-us/articles/360043076292-Copyright-overview

Table 1: Cultural representation of PhysGame dataset based on the continent of the game publisher.

| Region | Asia | Europe | Africa | America | Oceania |
|---|---|---|---|---|---|
| **proportion** | 29% | 27% | 2% | 37% | 5% |

Table 2: Distributions of game genres in PhysGame.

| Genres | Typical Games | Proportion |
|---|---|---|
| combat-themed games | Call of Duty, PUBG | 21% |
| platformers | Super Mario Bros, Hollow Knight | 12% |
| role-playing game | The Elder Scrolls V: Skyrim | 33% |
| action-adventure | The Legend of Zelda | 28% |
| strategy games | StarCraft II | 6% |

Experimental results demonstrate that: 1) Our PhysGame yields consistent improvements across different model sizes (3B/4B/7B/8B); 2) It enhances performance for diverse model families (Qwen, InternVL, and LLaVA series), demonstrating the strong generalization ability of the proposed physical glitch oriented tuning.

**Reliability of metadata.** We provide additional clarification about the potential noisiness of metadata (*i.e.*, Reddit post titles).

- **Metadata is only an auxiliary signal rather than a primary supervision source:** The QA pairs are generated by GPT-4o primarily based on the visual content of the video. Metadata is used only to guide the model toward the general direction of physical anomalies, not to supply the answer itself. When inconsistencies occur, GPT-4o naturally prioritizes the video evidence, which reduces the likelihood of hallucinations caused by noisy titles.

- **We incorporate quality control mechanisms to mitigate noisy metadata.** Our pipeline includes automated semantic consistency checks that filter out titles that are clearly unrelated to the video content. As shown in Table 2 of the main paper, incorporating metadata leads to a substantially higher QA accuracy (91% vs 64% without metadata), indicating that metadata generally enhances robustness rather than introducing additional noise.

- **Overall robustness is preserved despite potential noise in Reddit titles.** With appropriate prompt design and metadata filtering, noisy titles do not hinder the data generation process. Instead, metadata serves as a useful auxiliary cue that directs the model toward relevant physical phenomena and enhances the overall quality of the generated QA pairs.

**Qualitative examples of error types.** Figure 1 presents qualitative examples illustrating the three error types defined in our taxonomy: perception error, lack of knowledge, and reasoning error. In Figure 1a, the model misperceives the visual event itself, incorrectly identifying the skateboarding action as extreme high bungee jump activity. In Figure 1b, the model correctly identifies the helicopter in the scene but misinterprets the actual glitch, which is the excessively large bank angle during its maneuver, as a collision-detection issue related to ground contact, demonstrating a lack of physics knowledge. In Figure 1c, the model correctly identifies the physics glitch that a car suddenly collides with the player's vehicle. However, it mistakenly attributes this anomaly to the game's atmosphere and buoyancy model, which is irrelevant to the ground-truth collision-physics error.

**Discussions of potential biases.** To mitigate the dataset bias, we conducted a distribution analysis from the aspects of cultural representation, game genre diversity, player demographics, and harmful assumptions in the QA pairs. Specifically, we sampled 3,000 videos from our PhysGame dataset, and the audit results are as follows:

- **Cultural representation:** In Table 1, we categorized the games based on the continent of the game publisher as follows. The results show that our PhysGame dataset includes games developed both in the

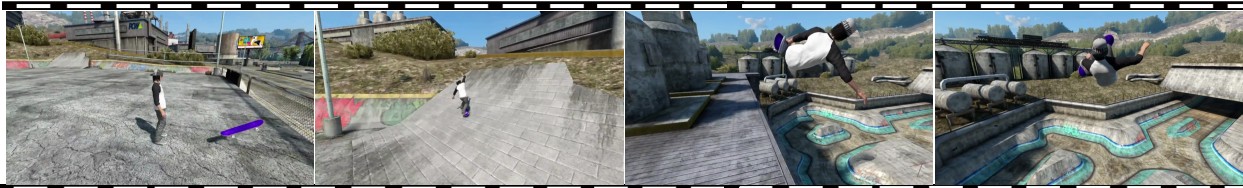

**Qwen2.5-VL-7B:** This video documents the character engaging in an extreme high bungee jump activity. He launched off the high incline, was shot out like an object from a catapult, and then spun and tumbled completely out of control in the air. This suggests he was attempting an incredibly difficult gymnastics aerial flip but misjudged the launch force, causing him to drastically overshoot the intended landing zone.

**Ground-truth:** The video shows a severe physics glitch where the **skateboarder** suddenly enters a ragdoll state and is catapulted immense distances at an unnatural speed upon hitting an inclined ramp. This "super launch" effect results from a **collision system error**, likely caused by hitting a specific geometric seam that triggers a physics engine conflict and dramatically escalates the character's velocity vector.

(a)

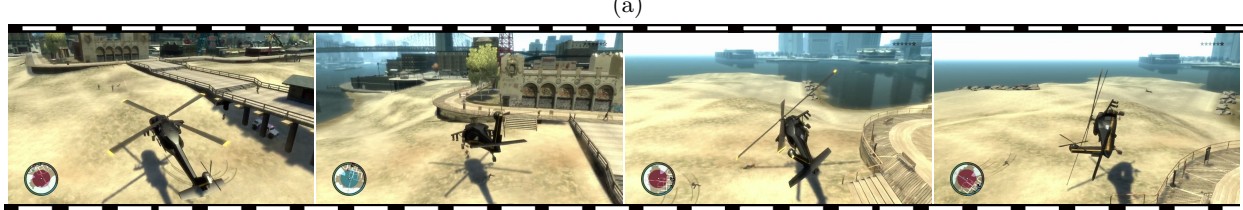

**Qwen2.5-VL-7B:** The video illustrates a collision-detection issue. When the helicopter attempts to land on the sandy shore near the water, it begins to lift upward before actually touching the ground. This suggests that the game's collision system may be prematurely or incorrectly identifying a portion of the shoreline as a solid surface.

**Ground-truth:** The fuselage exhibited an excessive bank angle during the helicopter's ascending and turning maneuver, which is inconsistent with realistic aerodynamic and flight dynamics characteristics.

(b)

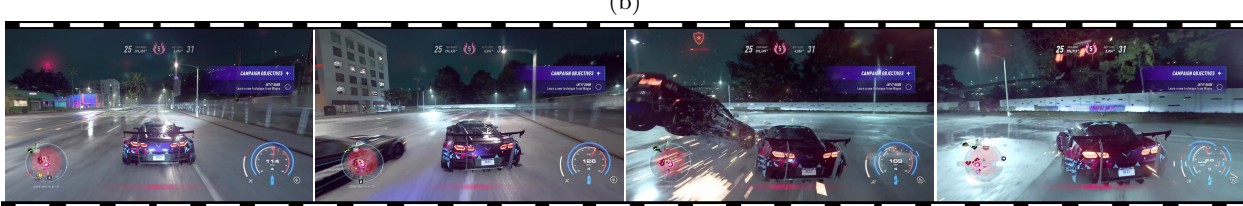

**Qwen2.5-VL-7B:** The video captures a severe physics glitch where a car dramatically "catapults" from the left, striking the player's vehicle at an unnatural speed and trajectory. This glitch is due to the game's **atmosphere and buoyancy model**. The system calculated an excessive upward thrust, similar to a depth charge exploding under water, pushing the lightened vehicle up.

**Ground-truth:** A police car suddenly catapults into the frame from the left, striking the player at an unnatural speed, indicating a severe physics glitch. This phenomenon is likely caused by a **physics engine error**. Specifically, the police car, when spawned or reset outside the player's view, may have **received an erroneous, massive kinetic force** (perhaps from a calculation overflow or a collision with an unloaded object), causing it to be instantly projected across the map at an impossible velocity and trajectory toward the player.

(c)

Figure 1: Qualitative examples of the three error types. (a) Perception error; (b) Lack of knowledge; and (c) Reasoning error.

West (*e.g.*, America) and the East (*e.g.*, Asia), with the majority of titles coming from American and Asian companies.

- **Game genres:** PhysGame contains a diverse set of game genres, including combat-themed games, platformers, role-playing games, action-adventure, and strategy games. We compiled statistics for each game genre and the typical games in Table 2.

- **Player demographics:** Since Reddit does not provide specific player identity information, and collecting such data may raise privacy concerns, we did not attempt to analyze or infer player demographics.

- **Harmful assumptions:** We conducted a bias audit on the instruction-tuning questions, checking for stereotypical phrasing, cultural bias, racial bias, age bias, and related issues. Our analysis found no

Table 3: Experimental results based on additional baselines.

| Models | GameBench | PhysBench | MMVU | MVBench | Video-MME | LongVideoBench |
|---|---|---|---|---|---|---|
| Qwen2.5VL-3B | 40.7 | 25.3 | 46.6 | 66.6 | 58.1 | 55.8 |
| *w/* PhysGame | 44.5 | 27.8 | 51.2 | 68.0 | 59.4 | 57.2 |
| Δ | +3.8 | +2.5 | +4.6 | +1.4 | +1.3 | +1.4 |
| Qwen2-VL-7B | 37.5 | 46.6 | 42.1 | 64.5 | 58.0 | 55.3 |
| *w/* PhysGame | 43.8 | 49.1 | 48.6 | 66.4 | 59.3 | 56.1 |
| Δ | +6.3 | +2.5 | +6.5 | +1.9 | +1.3 | +0.8 |
| Qwen2.5-VL-7B | 44.4 | 45.9 | 48.8 | 66.0 | 64.7 | 56.0 |
| *w/* PhysGame | 48.1 | 47.6 | 50.5 | 67.3 | 65.8 | 58.3 |
| Δ | +3.7 | +1.7 | +1.7 | +1.3 | +1.1 | +2.3 |
| InternVL2.5-4B | 46.5 | 25.0 | 48.7 | 71.5 | 60.8 | 53.6 |
| *w/* PhysGame | 50.1 | 27.5 | 52.9 | 72.6 | 61.9 | 55.8 |
| Δ | +3.6 | +2.5 | +4.2 | +1.1 | +1.1 | +2.2 |
| InternVL2.5-8B | 38.6 | 43.9 | 41.1 | 72.0 | 64.0 | 57.2 |
| *w/* PhysGame | 48.0 | 46.0 | 49.6 | 72.8 | 64.2 | 59.5 |
| Δ | 9.4 | +2.1 | +8.5 | +0.8 | +0.2 | +2.3 |
| LLaVA-Video-7B | 50.9 | 25.3 | 46.6 | 66.5 | 58.1 | 56.0 |
| *w/* PhysGame | 54.2 | 27.6 | 49.8 | 67.7 | 59.3 | 57.5 |
| Δ | +3.3 | +2.3 | +3.2 | +1.2 | +1.2 | +1.5 |
| LLaVA-OneVision1.5-8B | 44.2 | 24.2 | 47.2 | 52.9 | 60.5 | 56.7 |
| *w/* PhysGame | 47.8 | 26.5 | 50.1 | 54.3 | 61.7 | 58.4 |
| Δ | +3.6 | +2.3 | +2.9 | +1.4 | +1.2 | +1.7 |

Table 4: **Prompt** for instruction-tuning data generation in PhysGame.

> "role": "system"
> You are an AI visual assistant, and you are seeing a video and a title as a hint. Watch the video carefully and analyze the events and object movements, focusing on any inconsistencies with physical laws. Please design a conversation between you and the person asking about the game description and the glitch especially.
> Example questions:
> What is the description of the glitch observed in the video?
> Are there any abnormalities present in the videos?
> Please provide a description of the video content.
>
> "role": "user"
> Title of the video: "{`title`}". This hint might not be accurate. Your analysis should be based primarily on your own observations and understanding of the video and do not imply the title in any of your generation. Please directly design questions like the example and answer them in detail. Ensure that all descriptions are at the video level, do not refer to "images" or "frames".

evidence of such concerns, indicating that GPT-4o demonstrates strong safety control and consideration for ethical and unbiased language generation.

**Prompt Designs.** The prompts used for PhysGame data generation are shown in Table 4. The evaluation prompts are available in Table 5.

# References

Xiang An, Yin Xie, Kaicheng Yang, Wenkang Zhang, Xiuwei Zhao, Zheng Cheng, Yirui Wang, Songcen Xu, Changrui Chen, Chunsheng Wu, et al. Llava-onevision-1.5: Fully open framework for democratized multimodal training. *arXiv preprint arXiv:2509.23661*, 2025.

Table 5: **Evaluation prompts.**

**Evaluation prompts for GameBench:**
Watch the video carefully and analyze the events and object movements, focusing on any inconsistencies with physical laws. Identify and highlight instances where the behavior deviates from expected real-world physics, and select the most accurate option to describe the detected glitch.
Answer with the option letter from the given choices directly.
**Evaluation prompt for PhysBench:**
Answer with the option's letter from the given choices directly. You can only answer one letter from A, B, C, or D.
**Evaluation prompt for MMVU:**
Do not generate any intermediate reasoning process. Answer directly with the option letter from the given choices.
**Evaluation prompt for MVBench:**
Only give the best option.
**Evaluation prompt for Video-MME:**
Select the best answer to the following multiple-choice question based on the video and the subtitles. Respond with only the letter (A, B, C, or D) of the correct option.
**Evaluation prompt for LongVideoBench:**
Answer with the option's letter from the given choices directly.

Shuai Bai, Keqin Chen, Xuejing Liu, Jialin Wang, Wenbin Ge, Sibo Song, Kai Dang, Peng Wang, Shijie Wang, Jun Tang, et al. Qwen2.5-vl technical report. *arXiv preprint arXiv:2502.13923*, 2025.

Zhe Chen, Weiyun Wang, Yue Cao, Yangzhou Liu, Zhangwei Gao, Erfei Cui, Jinguo Zhu, Shenglong Ye, Hao Tian, Zhaoyang Liu, et al. Expanding performance boundaries of open-source multimodal models with model, data, and test-time scaling. *arXiv preprint arXiv:2412.05271*, 2024.

Peng Wang, Shuai Bai, Sinan Tan, Shijie Wang, Zhihao Fan, Jinze Bai, Keqin Chen, Xuejing Liu, Jialin Wang, Wenbin Ge, et al. Qwen2-vl: Enhancing vision-language model's perception of the world at any resolution. *arXiv preprint arXiv:2409.12191*, 2024.

Yuanhan Zhang, Jinming Wu, Wei Li, Bo Li, Zejun Ma, Ziwei Liu, and Chunyuan Li. Video instruction tuning with synthetic data. *arXiv preprint arXiv:2410.02713*, 2024.