# OpenReview forum: "Order from Chaos: Physical World Understanding from Glitchy Gameplay Videos"
_TMLR — Accepted by TMLR_

### Review · Reviewer_sbHr · 2025-10-31

**Summary Of Contributions:**

This paper introduces a novel paradigm for enhancing the physical world understanding of multi-modal large language models (MLLMs), centered on the idea of "Order from Chaos." The authors posit that by learning to identify "glitches" (bugs or anomalies) in gameplay videos that violate physical laws, models can gain a deeper understanding of real-world physical rules.

The primary contributions are two-fold:

- **PhysGame Dataset**: A large-scale instruction-tuning dataset containing 140,057 glitch-centric question-answer pairs. This dataset covers five physical domains (e.g., mechanics, optics) and was built using a novel "meta-information-guided prompting strategy" that leverages metadata like video titles to guide high-quality QA generation .

- **GameBench Benchmark**: An expert-annotated evaluation benchmark with 880 glitch-identified gameplay videos. It is designed to evaluate a model's ability to detect physically implausible events and was constructed with rigorous quality control, including LLM-assisted inspection and human cross-checking .

Key Strengths:

- **Novel and Scalable Approach**: Using glitch videos as a data source is a highly novel perspective. It cleverly bypasses the limitations of real-world data (high annotation cost) and traditional synthetic data (lacking realism and diversity). This approach is highly scalable due to the abundance of online gameplay footage.

- **Convincing Transferability**: The paper's strongest arguments are its "Game2Real" and "Game2General" transferability. Experiments compellingly demonstrate that models trained on game glitches show significant improvement in understanding real-world physics (e.g., on PhysBench) and general video understanding (e.g., on MVBench) .

- **Rigorous Dataset Construction and Evaluation**: The authors not only propose a training set but also a companion evaluation benchmark (GameBench). The construction method for PhysGame (meta-information-guided) is proven effective (QA accuracy boosted from 64% to 91% ). GameBench's construction is also well-considered, especially its distractor design , making it a challenging benchmark.

Key Weaknesses:

- **Dependence on General-Purpose Data**: The ablation study (Table 6) shows that training only on PhysGame (140K/0K) actually performs worse (45.1%) than the baseline using only general video data (0K/160K) (45.9%). This indicates a need for general data to prevent overfitting to the gameplay domain, slightly weakening the strength of glitch data as a standalone source and highlighting the importance of the mixing ratio.

**Audience:**

Yes

**Audience Explanation:**

Yes, a broad audience at TMLR would be interested in this paper for several reasons:

- **Multimodal and Video Understanding Community**: The paper addresses a core challenge in AI—physical world understanding. This is highly relevant to researchers in MLLMs, video analysis, and commonsense reasoning.

- **Data-Centric AI Community**: The paper presents a novel, low-cost, and highly scalable method for dataset construction. The insight of using "glitches" and metadata to auto-generate high-quality instruction-tuning data is extremely valuable for researchers focused on data curation and management.

- **Sim-to-Real Community**: The "Game2Real" transferability finding is particularly compelling. Bridging the sim-to-real domain gap is a long-standing problem. This paper offers a new path: instead of striving for perfect simulation, learn from the simulation's flaws. This opens new lines of thought for domain adaptation and generalization research.

**Broader Impact Concerns:**

I have no major ethical or broader impact concerns regarding this work.

- The research aims to advance the fundamental understanding of physical commonsense in AI, a goal of significant scientific merit.

- The dataset is sourced from a public Reddit forum (GamePhysics subreddit), which is standard practice in MLLM research and does not involve sensitive personally identifiable information.

- The technology does not have an obvious dual-use case or direct negative societal application.

**Claims And Evidence:**

Yes

**Claims Explanation:**

Yes, the paper's central claims are supported by sufficient and convincing evidence:

- **"Game2Real" Transferability Claim**: This is the boldest claim. The authors support it by experimenting on two benchmarks based on real-world videos (PhysBench and MMVU) . As shown in Table 4, models fine-tuned on PhysGame (e.g., Qwen2-VL-7B) achieved a 2.5% absolute gain on PhysBench (from 46.6% to 49.1%) and a 6.5% gain on MMVU (from 42.1% to 48.6%). InternVL2.5-8B also saw a significant 8.5% gain on MMVU.

- **"Game2General" Transferability Claim**: This claim is also well-supported by data in Table 5. Models show consistent performance boosts across multiple general video understanding benchmarks (MVBench, Video-MME, LongVideoBench) . For instance, Qwen-2-VL-7B improved by 1.9% on MVBench.

- **Data Construction Method Effectiveness**: The authors claim their "meta-information-guided prompting" is crucial for high-quality data. This is doubly verified: (a) Manual quality validation showed QA accuracy at 91% with meta-info vs. 64% without; (b) A quantitative ablation showed a 4.7% absolute performance gain on PhysBench from models trained with meta-info.

- **Value of the New Benchmark (GameBench)**: The authors claim GameBench is a challenging benchmark for physical reasoning. The results in Table 3 confirm this, as even SOTA proprietary models (like GPT-4o and Gemini 1.5-pro) only achieve 56.1% and 55.2% accuracy, respectively. This performance, far from human-level, proves the benchmark's utility and difficulty.

**Requested Changes:**

The paper is of high quality and well-argued. The following suggestions are intended to strengthen the work further and are not critical for acceptance:

- **(Strengthening) Deeper analysis of the data mixture ratio**: As mentioned, the ablation in Table 6 is very interesting . The finding that the 140K/0K (pure PhysGame) configuration performs poorly (45.1%) is attributed to domain bias. The paper would be stronger with more analysis on this "catastrophic forgetting" or domain bias phenomenon. For instance, why are the gains from 80K/80K (47.3%) to 140K/20K (47.6%)  so marginal? A deeper discussion on the interplay between the PhysGame data and general-purpose data (LLaVA-Hound) would be valuable.

- **(Strengthening) Qualitative example of a "Lack of Knowledge" error**: The error analysis in Figure 6 is excellent, identifying "Lack of Knowledge" as the primary error type (79-81%) for SOTA models. It would be helpful to provide one qualitative example of what this error looks like. For instance, does the model correctly perceive the visual phenomenon (e.g., "car flies into the air") but fail to identify the physical principle being violated (e.g., "gravity"), or does it misattribute it to another, incorrect physical principle? This would give readers a more concrete understanding of this dominant error category.

- **(Minor Strengthening) Clarify GPT-4o's role in evaluation**: The authors mention using GPT-4o for data generation and as a filter in GameBench's quality control (to remove questions answerable by text alone) . In Table 3, GPT-4o is also an evaluated model, scoring 56.1%. It would be worth adding a single sentence in the Section 4.2 discussion of this result, reminding the reader that this 56.1% score is achieved after the samples GPT-4o found "easy" were already filtered out. This further highlights the challenging nature of the GameBench benchmark. (While this is mentioned in Sec 3.2, a reminder in the results section would be impactful.)

---

> ### Author Response · Authors · 2025-12-14
> **Round-1 Rebuttal (1/2)**
>
> We sincerely thank all the reviewers for their insightful and constructive feedback. We provide our detailed, point-by-point responses to their comments as follows.
>
> > **sbHr-Q1**: Dependence on General-Purpose Data: The ablation study (Table 6) shows that training only on PhysGame (140K/0K) actually performs worse (45.1%) than the baseline using only general video data (0K/160K) (45.9%). This indicates a need for general data to prevent overfitting to the gameplay domain, slightly weakening the strength of glitch data as a standalone source and highlighting the importance of the mixing ratio.
>
> **A**: We thank the reviewer for highlighting this point. Regarding the observation that training only on PhysGame (140K/0K) performs slightly worse than using only general-purpose video data (0K/160K), we would like to clarify that these two settings differ in total data volume: PhysGame contains 140K samples, whereas the general-purpose dataset contains 160K. Therefore, this difference does not necessarily indicate that PhysGame provides weaker supervision, but may instead reflect the effect of training sample quantity.
>
> Moreover, our objective is not to show that PhysGame can replace general-purpose video data entirely, but rather to demonstrate that it provides specialized and complementary physical reasoning signals. As shown in Table 6 of the revised manuscript, combining PhysGame with a small amount of general video data (20K LLaVA-Hound) allows the model to retain broad visual diversity while simultaneously gaining stronger physical understanding, illustrating the complementary roles of the two data sources.
>
>
> > **sbHr-Q2**: Deeper analysis of the data mixture ratio: As mentioned, the ablation in Table 6 is very interesting . The finding that the 140K/0K (pure PhysGame) configuration performs poorly (45.1%) is attributed to domain bias. The paper would be stronger with more analysis on this "catastrophic forgetting" or domain bias phenomenon. For instance, why are the gains from 80K/80K (47.3%) to 140K/20K (47.6%) so marginal? A deeper discussion on the interplay between the PhysGame data and general-purpose data (LLaVA-Hound) would be valuable.
>
> **A**: Thank you for the insightful suggestion regarding a deeper analysis of the data mixture ratio. We have expanded our discussion in the revised manuscript to address the interesting observations more thoroughly, particularly concerning domain bias and diminishing returns.
>
> The performance of the pure PhysGame configuration (140K/0K) is slightly lower than that of the pure general-data configuration (0K/160K) primarily due to two factors. First, the difference in total training samples (140K vs. 160K) gives the general-data setting a scale advantage in learning diverse visual representations. Second, training solely on PhysGame, which is visually centered on gameplay scenes, can induce a mild domain bias, slightly hindering generalization to the broader visual distributions found in real-world benchmarks like PhysBench.
>
> Regarding the marginal gain when increasing PhysGame data from 80K to 140K while reducing general data from 80K to 20K (47.3% to 47.6%), we provide a two-fold explanation. First, this aligns with the principle of diminishing returns: after the model has absorbed the core physical reasoning patterns from the initial 80K PhysGame samples, additional data yields progressively smaller improvements. Second, benchmarks such as PhysBench inherently require strong general visual understanding (e.g., object recognition, scene relationships), capabilities that are better sustained by a foundational amount of general-purpose data like LLaVA-Hound. Therefore, significantly reducing the general data portion limits further gains.
>
> In summary, our analysis highlights a complementary relationship: PhysGame provides high-density, structured signals for physical reasoning, while LLaVA-Hound preserves visual diversity and prevents catastrophic forgetting or domain overfitting. This interplay ensures robust generalization. We have incorporated this extended analysis into a new subsection (Section 4.4) of the revised manuscript.

---

> ### Author Response · Authors · 2025-12-14
> **Round-1 Rebuttal (2/2)**
>
> > **sbHr-Q3**: Qualitative example of a "Lack of Knowledge" error: The error analysis in Figure 6 is excellent, identifying "Lack of Knowledge" as the primary error type (79-81%) for SOTA models. It would be helpful to provide one qualitative example of what this error looks like. For instance, does the model correctly perceive the visual phenomenon (e.g., "car flies into the air") but fail to identify the physical principle being violated (e.g., "gravity"), or does it misattribute it to another, incorrect physical principle? This would give readers a more concrete understanding of this dominant error category.
>
> **A:** We thank the reviewer for the thoughtful suggestion. As defined in our paper, a lack of knowledge error refers to cases where the model correctly perceives the visual phenomenon but fails to recognize the underlying physical principle being violated. To make this category more concrete, we have added qualitative examples in the supplementary material. Specifically, in Figure 1(a) of the supplementary material, the model misperceives the visual event itself, incorrectly identifying the skateboarding action as the extreme high bungee jump activity. In Figure 1(b), the model correctly identifies the helicopter in the scene but misinterprets the actual glitch, which is the excessively large bank angle during its maneuver, as a collision-detection issue related to ground contact, demonstrating a lack of physics knowledge. In Figure 1(c), the model correctly identifies the physics glitch that a car suddenly collides with the player's vehicle. However, it mistakenly attributes this anomaly to the game's atmosphere and buoyancy model, which is irrelevant to the ground-truth collision-physics error.
>
>
> > **sbHr-Q4**: Clarify GPT-4o's role in evaluation: The authors mention using GPT-4o for data generation and as a filter in GameBench's quality control (to remove questions answerable by text alone) . In Table 3, GPT-4o is also an evaluated model, scoring 56.1%. It would be worth adding a single sentence in the Section 4.2 discussion of this result, reminding the reader that this 56.1% score is achieved after the samples GPT-4o found "easy" were already filtered out. This further highlights the challenging nature of the GameBench benchmark. (While this is mentioned in Sec 3.2, a reminder in the results section would be impactful.)
>
> **A**: Thank you for this valuable suggestion. We agree that explicitly reminding the reader of the filtering step in the results section would strengthen the narrative around the benchmark's difficulty. In the revised manuscript, we have added the following sentence to the discussion of the GameBench results in Section 4.2:
>
> As a reminder, the GPT-4o score of 56.1% is achieved on GameBench after filtering out questions that were found to be answerable by text alone (using GPT-4o itself during quality control), which further underscores the challenging nature of the benchmark.

---

### Review · Reviewer_xhzp · 2025-11-04

**Summary Of Contributions:**

The paper proposes learning physical-world understanding from glitchy gameplay videos. It introduces PhysGame, a large instruction-tuning dataset of glitch-centric QA pairs, built via a meta-information–guided prompting scheme that leverages video titles/descriptions for higher-quality QA generation. The authors also release GameBench, an expert-annotated benchmark of gameplay clips with multiple-choice questions targeting physical implausibilities. Experiments show consistent transfer: finetuning on PhysGame improves real-world physical reasoning  and general video understanding. These results suggest that gameplay anomalies offer a scalable supervision signal for multimodal physical understanding.


Strengths

1. Clear motivation and generally well-written exposition; the ''order from chaos'' idea is intuitive and compelling.

2. Solid empirical evidence: consistent gains on real-world physics (PhysBench), general video benchmarks (MVBench/Video-MME), and the new GameBench.

3. Dataset design is scalable; construction details (taxonomy, metadata-guided prompting) are clearly documented.

Weaknesses

1. Cross-model analysis could be deepened (e.g., broader architectures/scaling trends or more systematic with/without PhysGame comparisons across tasks).

2. The ablation section’s setup and takeaways are not fully transparent; clearer descriptions of what’s varied (e.g., metadata usage, glitch types, filtering) and why would improve interpretability.

**Audience:**

Yes

**Audience Explanation:**

This work introduces an expert-annotated benchmark and demonstrates transfer to real-world physics and general video understanding. These elements make the findings relevant to researchers in video understanding, multimodal learning, and dataset design.

**Broader Impact Concerns:**

The paper lacks a Broader Impact statement and leaves key ethics points unclear. The dataset is built from Reddit gameplay videos without explicit disclosure of creator consent, licensing compliance, takedown procedures. Likely demographic/genre skews may introduce bias, with no distributional stats or disparate-impact analysis. Safety in high-risk uses (e.g., autonomous driving) and misuse risks (harmful content, deepfakes) are not addressed. Environmental costs of training and annotation subjectivity (guidelines, agreement, compensation) are also unreported.

**Claims And Evidence:**

Yes

**Claims Explanation:**

The paper shows that strong video understanding models struggle to achieve high accuracy on the proposed GameBench, indicating meaningful headroom and validating the benchmark’s challenge.

After fine-tuning on the PhysGame dataset, multiple model families exhibit performance improvements on GameBench, supporting the claim that PhysGame imparts useful physical reasoning rather than overfitting to a single architecture.

 Fine-tuned models also achieve higher accuracy on two real-world physics datasets, demonstrating transferability from gameplay videos to natural videos and directly supporting the core claim about real-world physical understanding.

Improvements on standard video-understanding benchmarks further suggest that PhysGame enhances models' general video comprehension, not only physics-specific judgments.

Overall, the evidence is coherent across benchmarks, models, and domains, making the submission’s claims convincing and well supported.

**Requested Changes:**

1. Broaden cross-model comparisons. Report with/without PhysGame across more model families and sizes (e.g., additional open-source baselines and parameter scales) to clarify architecture- and scale-robustness of the gains.

2. Quantify data-scaling effects. Provide a clear scaling curve (e.g., 0->40k->80k->140k PhysGame samples) on GameBench, PhysBench, and MVBench to test whether accuracy continues to improve with more PhysGame data. (Some ablations exist; please extend them to cover GameBench and general-video tasks explicitly.)

3. Clarify the ablation setup around LLaVA-Hound resampling. Make explicit whether resampling LLaVA-Hound contributes to the PhysBench gains. Include fixed-budget controls, seed-controlled resampling, and an ablation that isolates meta-information prompting vs. data-source mixing so readers can attribute improvements unambiguously.

---

> ### Author Response · Authors · 2025-12-14
> **Round-1 Rebuttal (1/4)**
>
> We sincerely thank all the reviewers for their insightful and constructive feedback. We provide our detailed, point-by-point responses to their comments as follows.
>
> > **xhzp-Q1**: Broaden cross-model comparisons. Report with/without PhysGame across more model families and sizes (e.g., additional open-source baselines and parameter scales) to clarify architecture- and scale-robustness of the gains.
>
>
> **A**: We thank the reviewer for the suggestion to broaden cross-model comparisons. In the main paper, we already evaluate PhysGame on three representative open-source MLLMs (Qwen2-VL-7B, Qwen2.5-VL-7B, InternVL2.5-8B), demonstrating consistent improvements on both Game2Real and Game2General transfer tasks. To further strengthen the claim that our approach is robust across architectures and model scales, we additionally evaluate four more models of different families and parameter sizes: Qwen2.5-VL-3B, InternVL2.5-4B, LLaVA-Video-7B, and LLaVA-OneVision-1.5-8B.
>
> The results with and without PhysGame finetuning on GameBench, PhysBench, MMVU, MVBench, Video-MME, and LongVideoBench are summarized below.
>
> |  Models | GameBench | PhysBench | MMVU | MVBench | Video-MME |  LongVideoBench |
> | ----------- | :-----------: | :-----------: | :-----------: | :-----------: | :-----------: | :-----------: |
> | Qwen2.5VL-3B | 40.7 | 25.3 | 46.6 | 66.6 | 58.1 | 55.8 |
> | w/ PhysGame  | 44.5 | 27.8 | 51.2	| 68.0 | 59.4 | 57.2 |
> |  $\Delta$    | +3.8 | +2.5 | +4.6	| +1.4 | +1.3 | +1.4 |
> | Qwen2-VL-7B |  37.5 | 46.6 | 42.1 | 64.5 | 58.0 | 55.3 |
> | w/ PhysGame  | 43.8 | 49.1 | 48.6 | 66.4 | 59.3 | 56.1 |
> |  $\Delta$    | +6.3 | +2.5 | +6.5 | +1.9 | +1.3 | +0.8 |
> | Qwen2.5-VL-7B | 44.4 | 45.9 | 48.8 | 66.0 | 64.7 | 56.0 |
> | w/ PhysGame  | 48.1 | 47.6 | 50.5 | 67.3 | 65.8 | 58.3 |
> |  $\Delta$    | +3.7 | +1.7 | +1.7 | +1.3 | +1.1 | +2.3 |
> | InternVL2.5-4B |  46.5 | 25.0 | 48.7 | 71.5 | 60.8 | 53.6 |
> | w/ PhysGame  | 50.1 |27.5	| 52.9	| 72.6	| 61.9	| 55.8 |
> |  $\Delta$    | +3.6 |	+2.5 |	+4.2 | +1.1	| +1.1	| +2.2 |
> | InternVL2.5-8B | 38.6 |  43.9 | 41.1 | 72.0 | 64.0 | 57.2 |
> | w/ PhysGame  | 48.0 | 46.0 | 49.6 | 72.8 | 64.2 | 59.5 |
> | $\Delta$  | +9.4 | +2.1 | +8.5 | +0.8 | +0.2 | +2.3 |
> | LLaVA-Video-7B  | 50.9  | 25.3 | 46.6 | 66.5 | 58.1 | 56.0 |
> | w/ PhysGame  | 54.2 | 27.6 | 49.8 | 67.7	| 59.3	| 57.5 |
> |  $\Delta$    | +3.3 |	+2.3 | +3.2	| +1.2	| +1.2	| +1.5 |
> | LLaVA-OneVision1.5-8B | 44.2 | 24.2 | 47.2 | 52.9 | 60.5 | 56.7 |
> | w/ PhysGame  | 47.8	| 26.5 | 50.1 | 54.3 | 61.7	| 58.4 |
> |  $\Delta$    | +3.6	| +2.3 |+2.9  | +1.4 | +1.2	| +1.7 |
>
> Experimental results demonstrate that: 1) Our PhysGame yields consistent improvements across different model sizes (3B/4B/7B/8B); 2) It enhances performance for diverse model families (Qwen, InternVL, and LLaVA series), demonstrating the strong generalization ability of the proposed physical glitch oriented tuning.
>
>
> > **xhzp-Q2**: Quantify data-scaling effects. Provide a clear scaling curve (e.g., 0->40k->80k->140k PhysGame samples) on GameBench, PhysBench, and MVBench to test whether accuracy continues to improve with more PhysGame data. (Some ablations exist; please extend them to cover GameBench and general-video tasks explicitly.)
>
> **A**: We thank the reviewer for the constructive suggestion to quantify data-scaling behavior more explicitly. In response, we conducted additional experiments using Qwen2.5-VL-7B fine-tuned with different amounts of PhysGame data (0K, 40K, 80K, 140K) to analyze scaling trends across GameBench, PhysBench, MMVU, MVBench and Video-MME. As noted in Section 4.1 of the main paper, for real-world and general-video tasks (PhysBench, MMVU, MVBench, Video-MME, LongVideoBench), we mix PhysGame with 20K LLaVA-Hound samples to maintain domain balance. The experimental results are as follows and the corresponding data-scaling curves are demonstrated in Figure 5 of the revised manuscript.
>
> |  Data Size | GameBench | PhysBench | MMVU | MVBench | Video-MME | LongVideoBench |
> | ----------- | :-----------: | :-----------: | :-----------: | :-----------:|:-----------:|:-----------:|
> | 0K        | 44.4 | 45.9 |  49.0 |  66.5 | 65.1 | 56.2 |
> | 40K       | 45.9 | 46.8 |  49.5 |  66.5 | 65.1 | 56.8 |
> | 80K       | 47.0 | 47.2 |  50.0 |  66.8 | 65.4 | 57.6 |
> | 140K      | 48.1 | 47.6 | 50.5 | 67.3 |  65.8 |  58.3 |
>
> Across all six benchmarks, performance consistently improves as more PhysGame data is used. The gains are monotonic and do not plateau even at 140K samples, indicating that the dataset’s scaling potential has not been saturated. This trend validates the scalability of PhysGame and suggests that further expansion could yield additional improvements.

---

> ### Author Response · Authors · 2025-12-14
> **Round-1 Rebuttal (2/4)**
>
> > **xhzp-Q3**: Clarify the ablation setup around LLaVA-Hound resampling. Make explicit whether resampling LLaVA-Hound contributes to the PhysBench gains. Include fixed-budget controls, seed-controlled resampling, and an ablation that isolates meta-information prompting vs. data-source mixing so readers can attribute improvements unambiguously.
>
> **A**: Thank you for the insightful comment regarding the need for a clearer ablation setup.
>
> We acknowledge that the original description of the LLaVA-Hound resampling was not explicit enough. We make the revisions as follows:
>
> **Fixed-budget and seed-controlled resampling:** As revised in Section 4.1 of the main paper, we now fix the total training budget to 160K samples and systematically vary the mixing ratio between PhysGame (PG) and LLaVA-Hound (LH) as [0K/160K, 40K/120K, 80K/80K, 140K/20K]. A fixed random seed ensures the same LH subset is used across runs, eliminating sampling randomness.
>
>
> We have conducted additional controlled experiments to isolate the contributions and will update the manuscript accordingly:
>
> **Isolating meta-information prompting vs. data mixing:** We designed two ablations using Qwen2.5-VL-7B: 1) _w/o_ meta-info: We generate a variant of PhysGame without using video titles during prompting, then fine-tune with 140K of this data and 20K LH; 2) _w/o_ mixture: We fine-tune using only PhysGame (140K) without any LH data. The results are summarized below:
>
>
> |  Method | PhysBench | MMVU | MVBench | Video-MME | LongVideoBench |
> | ----------- | :-----------: | :-----------: | :-----------: | :-----------:|:-----------:|
> | Baseline | 45.9 |  48.8 |  66.0 | 64.7 | 56.0 |
> | Full | **47.6** | **50.5** | **67.3** |  **65.8** |  **58.3** |
> | _w/o_ meta-info | 42.9 | 46.7 | 65.2 | 63.5 | 55.2 |
> | _w/o_ mixture |  45.1 | 49.2 | 66.7 | 65.1 | 57.4 |
>
> These results show that meta-information prompting emerges as the primary contributor, delivering substantial gains (e.g., +4.7% on PhysBench) over the variant without it. Data mixing also plays a clear role, particularly in enhancing generalization to real-world benchmarks (e.g., +2.5% on PhysBench compared to using only PhysGame). The combination of both strategies yields the best overall performance.
>
> We have incorporated the above experimental setup, results, and analysis into Section 4.4 of the revised manuscript.

---

> ### Author Response · Authors · 2025-12-14
> **Round-1 Rebuttal (3/4)**
>
> > **xhzp-Q4**: Broader Impact Concerns. The paper lacks a Broader Impact statement and leaves key ethics points unclear. The dataset is built from Reddit gameplay videos without explicit disclosure of creator consent, licensing compliance, takedown procedures. Likely demographic/genre skews may introduce bias, with no distributional stats or disparate-impact analysis. Safety in high-risk uses (e.g., autonomous driving) and misuse risks (harmful content, deepfakes) are not addressed. Environmental costs of training and annotation subjectivity (guidelines, agreement, compensation) are also unreported.
>
> We appreciate the reviewer for raising the Broader Impact concerns. In response, we provide discussions on licensing, distribution bias, safety in high-risk uses, and environmental cost. Finally, we conclude with an updated broader impact statement.
>
> 1. License discussion
>
> We understand and appreciate the importance of responsible data use. To clarify, all videos used in our dataset are publicly available and shared by users on Reddit under subreddit communities that operate under Reddit’s _User Agreement_ and _Content Policy_. According to Reddit's terms, users grant Reddit **a broad license to host and redistribute uploaded content**, and its API **allows lawful academic access to public posts for research purposes**.
>
> To better align with ethical standards and community expectations, we will make **the following modifications** in our final version:
>
> - **Switch to link-only indexing**: We will update our data release to only index and compile links to the original publicly available videos and meta-data on Reddit, without hosting or redistributing any video content ourselves.
> - **Add opt-out mechanism and usage documentation**: We will additionally update our dataset release with an _opt-out_ mechanism, allowing content owners to request removal of their video references, and provide clear documentation of the dataset’s terms of use. The specific contents are as follows: ``We uphold the rights of individuals and copyright holders. If you are featured in any of our video annotations or hold copyright to a video and wish to have its annotation removed from our dataset, please reach out to us. We commit to reviewing your request promptly and taking suitable action.``
>
> 2. Distribution bias: We appreciate the feedback about potential distributional biases. Upon careful examination, we believe that the proposed PhysGame **does not exhibit significant bias** in terms of cultural representation, game genre diversity, player demographics, or harmful assumptions in the QA pairs. Specifically, we sampled 3,000 videos from our PhysGame dataset, and the audit results are as follows:
>
> - **Cultural representation**: We categorized the games based on the continent of the game publisher as follows. The results show that our PhysGame dataset includes games developed both in **the West (e.g., America) and the East (e.g., Asia)**, with the majority of titles coming from American and Asian companies.
>
> |  Region | Asia | Europe | Africa | America | Oceania |
> | ----------- | :-----------: | :-----------: | :-----------: | :-----------: | :-----------: |
> | proportion  |  29%  | 27% | 2% | 37% | 5% |
>
> - **Game genres**: PhysGame contains **a diverse set of game genres**, including combat-themed games, platformers, role-playing games, action-adventure, and strategy games. We compiled statistics for each game genre and the typical games as follows.
>
> |  Genres | Typical Games | Proportion |
> | ----------- | :-----------: | :-----------: |
> | combat-themed games | Call of Duty, PUBG | 21% |
> | platformers | Super Mario Bros, Hollow Knight | 12% |
> | role-playing game | The Elder Scrolls V: Skyrim | 33% |
> | action-adventure | The Legend of Zelda| 28% |
> | strategy games | StarCraft II | 6% |
>
> - **Player demographics**: Since Reddit does not provide specific player identity information, and collecting such data may raise privacy concerns, we **did not attempt to analyze or infer player demographics**.
>
> - **Harmful assumptions in the QA pairs**: We conducted a bias audit on the instruction-tuning questions, checking for **stereotypical phrasing, cultural bias, racial bias, age bias**, and related issues. Our analysis found **no evidence of such concerns**, indicating that GPT-4o demonstrates strong safety control and consideration for ethical and unbiased language generation.
>
> 3. **Safety in high-risk uses:** We thank the reviewer for raising this important point. We would like to clarify that our work does not target high-risk application domains such as autonomous driving, nor does it enable any human-facing decision-making system that could lead to safety-critical consequences. Instead, our contribution is a training dataset and benchmark derived exclusively from gameplay videos, focusing on improving models’ physical commonsense reasoning ability.

---

> ### Author Response · Authors · 2025-12-14
> **Round-1 Rebuttal (4/4)**
>
> 4. **Environmental costs:** We have reported the GPU training hours and estimated carbon emissions as follows. We experiment on 8 NVIDIA A100 GPUs, and the carbon footprint is estimated using publicly available tools such as ``ML CO2 calculator``. Specifically, the amount of emitted carbon is calculated as the product of power consumption, GPU-Hours, and the carbon produced based on the local power grid.
>
> |  Metric | Value |
> | ----------- | :-----------: |
> | Training GPU-Hours   |   28 hours |
> | Carbon Footprint	 | 4.84 kg eq. CO$_2$ |
>
>
> 5. **Broader Impact statement:**
> We summarize the above discussions in the Broader Impact statement in the supplementary material:
>
> This work uses publicly available gameplay videos that contain only synthetic and non-personal virtual environments to construct scalable datasets for studying physical reasoning in MLLMs. Since the data do not depict real individuals or communities, the dataset does not encode demographic or cultural biases, and we provide genre-level statistics for transparency. All content is accessed under Reddit's standard usage terms, and we release only metadata and source links rather than re-distributing the videos themselves. The method does not generate or manipulate real imagery and is not intended for use in high-risk domains such as autonomous driving or robotics, so misuse risks such as harmful content or deepfakes are minimal. Overall, this work aims to advance research on physical commonsense while maintaining ethical data governance and transparent reporting.

---

### Review · Reviewer_64BW · 2025-12-05

**Summary Of Contributions:**

The authors argue that identifying violations of physical laws is a scalable way to teach physical principles. They introduce PhysGame, an instruction-tuning dataset with over 140k QA pairs pulled from glitchy gameplay videos collected from Reddit, along with GameBench, a manually curated benchmark for evaluation. The authors use a meta-information-guided prompting strategy to generate high-quality training data. Experiments show that fine-tuning with PhysGame improves performance on gameplay benchmarks and also transfers effectively to real-world physical understanding and general video understanding tasks.

**Audience:**

Yes

**Audience Explanation:**

The ideas and validation around learning from glitches could further expand the scale of the dataset. The proposed PhysGame and GameBench will be valuable resources for researchers dedicated to physical world understanding tasks.

**Broader Impact Concerns:**

There is no explicit Broader Impact Statement present in the paper.

**Claims And Evidence:**

Yes

**Claims Explanation:**

The authors have conducted thorough comparison and ablation experiments to back up their claims: learning from "glitches" improves physical understanding, PhysGame boosts general video understanding, and metadata is key for generating high-quality data.

**Requested Changes:**

- This relies a lot on the quality of Reddit post titles. While the authors validated a sample, Reddit titles can sometimes be sarcastic, unrelated, or use slang that doesn't really describe the physical phenomenon accurately. If the metadata is noisy, the teacher model (GPT-4o) might generate hallucinated explanations. While the paper briefly touches on this, it could go deeper into how these potential issues are handled, particularly in terms of robustness. There’s also no real discussion about how potential biases in the data are addressed or how they impact the dataset’s overall reliability. More transparency here would be helpful.
- The paper mentions filtering out non-game elements, but it’s unclear how the pipeline distinguishes between intentional game mechanics that break physics (which shouldn’t be "corrected") and unintentional rendering glitches. It would be useful if the authors clarified the line between a “glitch” and “game logic” (for example, double-jumping or magic spells that are part of the game’s design).
- For GameBench, the distractor options are designed to include objects that actually appear in the video, which prevents the model from just matching patterns. It would be helpful if the authors explained whether these distractors were generated automatically or written manually. If they were auto-generated, how did the authors ensure they were "deceptive" enough to challenge the model?

---

> ### Author Response · Authors · 2025-12-14
> **Round-1 Rebuttal (1/2)**
>
> We sincerely thank all the reviewers for their insightful and constructive feedback. We provide our detailed, point-by-point responses to their comments as follows.
>
> > **64BW-Q1**: This relies a lot on the quality of Reddit post titles. While the authors validated a sample, Reddit titles can sometimes be sarcastic, unrelated, or use slang that doesn't really describe the physical phenomenon accurately. If the metadata is noisy, the teacher model (GPT-4o) might generate hallucinated explanations. While the paper briefly touches on this, it could go deeper into how these potential issues are handled, particularly in terms of robustness.
>
> **A**: We thank the reviewer for highlighting the concern regarding the potential noisiness of Reddit post titles. We provide additional clarification below:
>
> - **Metadata is only an auxiliary signal rather than a primary supervision source.** The QA pairs are generated by GPT-4o primarily based on the visual content of the video. Metadata is used only to guide the model toward the general direction of physical anomalies, not to supply the answer itself. When inconsistencies occur, GPT-4o naturally prioritizes the video evidence, which reduces the likelihood of hallucinations caused by noisy titles. The prompt for instruction-tuning data generation in PhysGame is available in the supplementary material.
>
> - **We incorporate quality control mechanisms to mitigate noisy metadata.** Our pipeline includes automated semantic consistency checks that filter out titles that are clearly unrelated to the video content. As shown in Table 2 of the main paper, incorporating metadata leads to a substantially higher QA accuracy (91% vs 64% without metadata), indicating that metadata generally enhances robustness rather than introducing additional noise.
>
> - **Overall robustness is preserved despite potential noise in Reddit titles.** With appropriate prompt design and metadata filtering, noisy titles do not hinder the data generation process. Instead, metadata serves as a useful auxiliary cue that directs the model toward relevant physical phenomena and enhances the overall quality of the generated QA pairs.
>
> We have incorporated the above discussions in the revised manuscript (cf. page #2 in the supplementary material).
>
> > **64BW-Q2**: There's also no real discussion about how potential biases in the data are addressed or how they impact the dataset’s overall reliability. More transparency here would be helpful.
>
> **A**: To mitigate the dataset bias, we conducted a distribution analysis from the aspects of cultural representation, game genre diversity, player demographics, and harmful assumptions in the QA pairs. Specifically, we sampled 3,000 videos from our PhysGame dataset, and the audit results are as follows:
>
> - **Cultural representation**: We categorized the games based on the continent of the game publisher as follows. The results show that our PhysGame dataset includes games developed both in **the West (e.g., America) and the East (e.g., Asia)**, with the majority of titles coming from American and Asian companies.
>
> |  Region | Asia | Europe | Africa | America | Oceania |
> | ----------- | :-----------: | :-----------: | :-----------: | :-----------: | :-----------: |
> | proportion  |  29%  | 27% | 2% | 37% | 5% |
>
> - **Game genres**: PhysGame contains **a diverse set of game genres**, including combat-themed games, platformers, role-playing games, action-adventure, and strategy games. We compiled statistics for each game genre and the typical games as follows.
>
> |  Genres | Typical Games | Proportion |
> | ----------- | :-----------: | :-----------: |
> | combat-themed games | Call of Duty, PUBG | 21% |
> | platformers | Super Mario Bros, Hollow Knight | 12% |
> | role-playing game | The Elder Scrolls V: Skyrim | 33% |
> | action-adventure | The Legend of Zelda| 28% |
> | strategy games | StarCraft II | 6% |
>
> - **Player demographics**: Since Reddit does not provide specific player identity information, and collecting such data may raise privacy concerns, we **did not attempt to analyze or infer player demographics**.
>
> - **Harmful assumptions in the QA pairs**: We conducted a bias audit on the instruction-tuning questions, checking for **stereotypical phrasing, cultural bias, racial bias, age bias**, and related issues. Our analysis found **no evidence of such concerns**, indicating that GPT-4o demonstrates strong safety control and consideration for ethical and unbiased language generation.
>
> All the above analyses have been added in the supplementary material.

---

> ### Author Response · Authors · 2025-12-14
> **Round-1 Rebuttal (2/2)**
>
> > **64BW-Q3**: The paper mentions filtering out non-game elements, but it's unclear how the pipeline distinguishes between intentional game mechanics that break physics (which shouldn't be "corrected") and unintentional rendering glitches. It would be useful if the authors clarified the line between a "glitch" and "game logic" (for example, double-jumping or magic spells that are part of the game’s design).
>
> **A**: We appreciate the reviewer's insightful question regarding the distinction between intentional game logics and unintentional physics glitches. This distinction is indeed crucial to the reliability of our datasets.
>
> For GameBench (Evaluation): Each video and its multiple-choice questions are **manually curated and cross-validated** by expert annotators. These experts are explicitly instructed to select only unintentional rendering or physics engine bugs while filtering out designed game mechanics (e.g., double jumps, magical spells). This ensures a clean, unambiguous testbed.
>
> For PhysGame (Training Data): While full manual review is infeasible at scale, our pipeline is designed to maximize thematic consistency:
>
> - **Source Control:** We source videos from the GamePhysics subreddit, a community specifically dedicated to sharing unintended physical bugs and anomalies.
>
> - **Metadata-guided prompting:** Our meta-information-guided prompting steers the LLM (GPT-4o) to focus its analysis on physical plausibility and visual inconsistencies described in the title, rather than on designed game logic.
>
> - **Visual Patterns:** Unintentional glitches typically present clear violations of visual or physical continuity, such as collision failures, sudden jumps in motion, unsupported hovering, or rendering artifacts. In contrast, intentional game logits show coherent visual design, consistent artistic style, and structured feedback signals that align with the game's world rules. These differences allow the prompting strategy to guide the model toward genuine physical anomalies rather than designed gameplay behaviors.
>
>
> > **64BW-Q4**: For GameBench, the distractor options are designed to include objects that actually appear in the video, which prevents the model from just matching patterns. It would be helpful if the authors explained whether these distractors were generated automatically or written manually. If they were auto-generated, how did the authors ensure they were "deceptive" enough to challenge the model?
>
> **A**: We thank the reviewer for raising this question. All QA pairs in GameBench, including the distractor options, are **manually written** by annotators rather than automatically generated. The annotators watched the full gameplay videos and intentionally selected distractors from objects or entities that actually appear in the scene but are semantically incorrect for the given question. This manual process ensures that each distractor is not only visually grounded but also plausible enough to deceive a model that relies on superficial correlations.

---

### Decision · Action_Editor_W5uw · 2026-01-13

**Recommendation:** Accept with minor revision

**Additional Comments:**

Reviewers suggested that the authors could further strengthen the reproducibility of the results by providing additional intermediate evidence, such as representative training/validation curves or limited variance statistics across runs. While the current experimental results already provide strong support for the paper’s claims, it would be beneficial to see whether the authors can address this suggestion with some lightweight additions, without making it a strict requirement.

**Audience:**

Yes

**Audience Explanation:**

The paper addresses a fundamental problem in physical reasoning with a novel and scalable approach, and its findings on learning physical laws would be of interest to TMLR audience.

**Claims And Evidence:**

Yes

**Claims Explanation:**

The paper’s claims are supported by accurate, convincing, and clearly presented evidence. The additional analyses and qualitative examples in the response further strengthen the empirical support and clarify the scope and limitations of the findings.